
# Pollution slightly enhances atmospheric cooling by low-level clouds in tropical West Africa

Valerian Hahn[1,2], Ralf Meerkötter[1], Christiane Voigt[1,2], Sonja Gisinger[1], Daniel Sauer[1], Valéry Catoire[3], Volker Dreiling[4], Hugh Coe[5], Cyrille Flamant[6], Stefan Kaufmann[1], Jonas Kleine[1], Peter Knippertz[7],
Manuel Moser[1,2], Phil Rosenberg[8], Hans Schlager[1], Alfons Schwarzenboeck[9], Jonathan Taylor[5]

[1]Institute of Atmospheric Physics, Deutsches Zentrum für Luft- und Raumfahrt (DLR), Oberpfaffenhofen, Germany
[2]Institute of Atmospheric Physics, University Mainz, Mainz, Germany
[3]Laboratoire de Physique et Chimie de l'Environnement et de l'Espace, CNRS-CNES-Université d'Orléans, France
[4]Flight Experiments, Deutsches Zentrum für Luft- und Raumfahrt (DLR), Oberpfaffenhofen, Germany
[5]School of Earth, Atmospheric and Environmental Sciences, University of Manchester, United Kingdom
[6]Laboratoire Atmosphères Milieux Observations Spatiales, Sorbonne Universités, Paris, France
[7]Institute of Meteorology and Climate Research, Karlsruhe Institute of Technology, 76128 Karlsruhe, Germany
[8]School of Earth and Environment, University of Leeds, Leeds, UK
[9]Laboratoire de Météorologie Physique, Université Blaise Pascal, CNRS, Clermont-Ferrand, France

*Correspondence to*: Valerian Hahn (Valerian.Hahn@dlr.de)

**Abstract.** Reflection of solar radiation by tropical low-level clouds has an important cooling effect on climate and leads to decreases in surface temperatures. Still, the effect of pollution on ubiquitous tropical continental low-level clouds and the investigation of the related impact on atmospheric cooling rates are poorly constrained by in-situ observations and modelling,
in particular during the West African summer monsoon season. Here, we present comprehensive in-situ measurements of microphysical properties of low-level clouds over tropical West Africa, measured with the DLR aircraft *Falcon 20* during the DACCIWA (Dynamics–Aerosol–Chemistry–Cloud Interactions in West Africa) campaign in June and July 2016. Clouds below 1800 meter altitude, identified as boundary layer clouds, were classified according to their carbon monoxide (CO) pollution level into pristine and less polluted clouds (CO < 135 ppbv) and polluted low-level clouds (CO > 155 ppbv) as
confirmed by the linear CO to accumulation aerosol correlation. Whereas slightly enhanced aerosol background levels from biomass burning were measured across the entire area, clouds with substantially enhanced aerosol levels were measured in the outflow of major coastal cities, as well as over rural conurbations in the hinterlands. Here we investigate the impact of pollution on cloud droplet number concentration and size during the West African Monsoon season. Our results show that the cloud droplet number concentration (CDNC) measured in the size range from 3 µm to 50 µm around noon increases by 35% in the
elevated aerosol outflow of coastal cities and conurbations with elevated aerosol loadings from median CDNC of 240 cm$^{-3}$ (52 cm$^{-3}$ to 501 cm$^{-3}$ interquartile range to 324 cm$^{-3}$ (60 cm$^{-3}$ to 740 cm$^{-3}$ interquartile range). Higher CDNC resulted in a 17% decrease in effective cloud droplet diameter from a median $d_{eff}$ of 14.8 µm to a $d_{eff}$ of 12.4 µm in polluted clouds. Radiative transfer simulations show a non-negligible influence of droplet number concentrations and particle sizes on the net radiative forcing at the top of atmosphere of -16.3 W m$^{-2}$ of the polluted with respect to the less polluted clouds and lead to a
change in instantaneous heating rates of -18 K day$^{-1}$ at top of the clouds at noon. It was found that the net radiative forcing at top of atmosphere accounts for only 2.6 % of the net forcing of the cloud-free reference case. Thus, polluted low-level clouds add only a relatively small contribution on top of the already exerted cooling by low-level clouds in view of a background atmosphere with elevated aerosol loading. Additionally, the occurrence of mid- and high-level cloud layers atop buffer this effect further, so that the net radiative forcing and instantaneous heating rate of low-level clouds turn out to be less sensitive
towards projected future increases in anthropogenic pollution in West Africa.



## 1 Introduction

During the summer monsoon, warm and humid conditions lead to the widespread occurrence of low-level clouds in tropical West Africa (Knippertz et al., 2015; Hill et al., 2018). The refraction and attenuation of solar radiation by the enhanced scattering cross section of these liquid clouds results in a strong surface cooling (Hill et al., 2018). The low-level clouds are

often obscured by higher clouds (Figure 1) in passive remote sensing data from satellite, which has to be considered when evaluating the cloud's effects on surface temperature and the tropospheric radiation budget. The daytime frequency of low-level cloud occurrence, either isolated or in combination with mid and high-level clouds accounts for 48 % of the observed cases (Hill et al., 2018). Therefore in-situ measurements are required to quantitatively assess their impact on the atmospheric radiation budget (Knippertz et al., 2015). The role of cloud's climate sensitivity is yet ambiguous and as such limits the

confidence in projected climate change from global climate models (Eyring et al., 2020). The fast and small-scale evolution of clouds results in the need to parameterise cloud properties in global climate models considering cloud effects. In-situ observations in rarely probed regions including tropical Africa are needed to better constrain cloud droplet number concentration (CDNC) and the effective droplet diameter (ED) in global climate models (i.e. Righi et al., 2020).

With an annual 6 % increase of its gross domestic product (Knippertz et al., 2017), sub-Saharan Africa's economy and socio-

economic system is undergoing major changes. The ongoing population growth, urbanisation and industrialisation leads to strong increases in emissions from industry and the transport sector in particular in the major cities, but also from domestic fires in urban and more rural areas (Liousse et al., 2014). Hence one of the motivations of the international DACCIWA project (Dynamics-Aerosol-Chemistry-Clouds Interactions in West Africa) was to quantify and better understand the effects of increased pollution levels within the boundary layer on low-level clouds over tropical West Africa (Knippertz et al., 2015).

The DACCIWA project combined large scale satellite observations and local ground-based measurements with detailed trace gas, aerosol, cloud and meteorology observations from three airborne platforms: the German *Deutsches Zentrum für Luft- und Raumfahrt (DLR)* Falcon 20, the French *Service des Avions Français Instrumentés pour la Recherche en Environnement (SAFIRE)* ATR 42, and the *British Antarctic Survey (BAS)* Twin Otter (Flamant et al., 2018b). The DACCIWA aircraft campaign took place from 29 June to 16 July 2016 and was based in Lomé, Togo, with survey flights over Togo, Benin, Ghana

and the Ivory Coast. Taylor et al. (2019) give a comprehensive overview on aerosol and cloud properties detected in that region during the Africa monsoon period. By using the measured aerosol composition and size distribution they investigate the dependence of the cloud microphysical properties on updraft and aerosol loading. Similarly, cloud formation processes and their dependence on updraft and aerosol has been investigated in low level clouds over the Western Atlantic (Kirschler et al., 2022) and in convection (e.g. Braga et al., 2017a, b and 2022). A comprehensive analysis of the aerosol composition measured

during the DACCIWA campaign is given by Haslett et al. (2019). They find a large contribution of biomass burning aerosol transported from the southern hemisphere on the large mode of the background aerosol distribution in West Africa, which acts as cloud condensation nuclei. Interactions of CCN from biomass burning aerosol and low-level clouds have also been studied by Painemal et al. (2014). Deroubaix et al. (2022) investigate the sensitivity of low-level clouds and precipitation to anthropogenic aerosol emissions in southern West Africa based on the cloud water content measured on a single day on

common transects of the 3 research aircraft between Lomé (Togo) and Savé (Benin) and simulations with a regional meteorology chemistry model (Baklanov et al., 2014; Menut et al., 2019). Another study by Pante et al. (2021) finds a correlation between increased anthropogenic aerosol emission and reduced rainfall. A comparison of inland and offshore aerosol and cloud distributions suggests a moderate impact of local emissions on cloud droplet number concentrations due to an enhanced biomass burning aerosol mode, which likely dampens the effect of additional aerosol emissions on cloud drop

number concentrations (Taylor et al., 2019).

Model simulations of the effects of changing aerosol emissions on the cloud fields (Deetz et al., 2018) suggest that the reduction of land-sea temperature gradient from increasingly hazy conditions led to a weakening of the monsoon flow and a delay in cloud breakup. Yearly variabilities in monsoon yields are analysed by Grist et al. (2002). Hill et al. (2018) statistically analysed



the radiative effects from 12 distinct cloud types over south West Africa during the summer period from June to September

using satellite data. A surface cooling of all indicated cloud types has been calculated from this data set. The study assigns

only a small radiative cooling of the atmosphere to low-level clouds.

Driven by the African monsoon, tropical low-level stratiform clouds developed during the night and early morning hours and

persisted throughout noon till they finally dissipated in the afternoon (van der Linden et al., 2015; Flamant et al., 2018a; Taylor

et al., 2019). The DACCIWA period was characterised by over 85 % coverage of low clouds, defined as clouds below 800 hPa

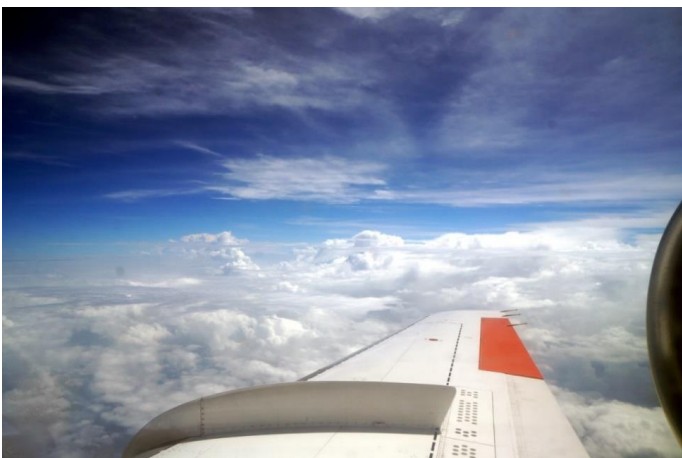


**Figure 1: Cloud situation as seen from the DLR Falcon research aircraft from the DACCIWA campaign with low-level cumuli as sequel of the nightly low-level stratus deck, convective clouds with significant vertical extent and various cirrus layers atop.**

pressure level (Knippertz et al., 2017), which corresponds roughly to 1800 m according to the standard atmosphere and thus

will be used as boundary layer height in this study. Boundary layer meteorology was dominated by a south westerly flow and

the free troposphere above by the African easterly jet (Knippertz et al., 2017). A sea breeze governed the wind patterns in the

coastal area in the early afternoon (Flamant et al. 2018b).

While Deroubaix et al. (2022) use a modelling approach to study the sensitivity of low-level clouds and precipitation, Taylor

et al. (2019) present a comprehensive statistical analysis of cloud properties from all 3 research aircraft and assess the cloud

droplet activation by a variation of local aerosol emission. Hill et al. (2018) analyse a cloud radiative effect from satellite

observations. In our study we calculate the radiative impact of inland continental low-level clouds in West Africa based on a

comprehensive data set of in-situ observations from the 12 measurement flights of the Falcon 20 research aircraft during the

DACCIWA airborne campaign. Additionally, we simulate how the increased anthropogenic pollution of low-level clouds

affects the radiation budget and in which direction such effects could change the local climate.

An overview of the instruments and the methodology is described in Section 2. In Sect. 3 and 4 the description of the

microphysical quantities in low-level clouds follows, with which in Sect. 5 radiative transfer calculations are carried out and

their results are presented.

## 2 Instrumentation and methods

The influence of aerosol loading on microphysical properties of low-level clouds is derived from in-situ measurements of

clouds with the Cloud and Aerosol Spectrometer CAS installed at a wing station of the Falcon 20 research aircraft. In addition,

the European Centre for Medium-Range Weather Forecasts' (ECMWF) integrated forecast system (IFS) provided the basis

for flight planning. Atmospheric state parameters from ECMWF Copernicus Atmosphere Monitoring Service (CAMS) were

used for radiative transfer calculations with the software package *libRadtran* (Mayer and Kylling, 2005). From these

calculations, we investigate the influence of an increase of cloud droplet number concentrations with decreasing droplet

diameter in low-level clouds on radiative forcing and atmospheric heating rates.



### 2.1 In-situ trace gas, aerosol and cloud instrumentation on the Falcon research aircraft

Particles and trace gases were measured with a set of well characterised instruments that have been deployed aboard the Falcon in previous flight campaigns (e.g. Voigt et al., 2010; Kleine et al., 2018; Voigt et al., 2022).

**The Cloud and Aerosol Spectrometer CAS**

The Cloud and Aerosol Spectrometer CAS (Baumgardner et al., 2001; Voigt et al., 2021) mounted on an inner underwing position of the Falcon was used to measure the cloud droplet number and the droplet size distribution in a size range between 0.5 µm to 50 µm. A photodetector senses forward scattered light from an annulus between 4 ° to 12 ° from particles that pass through a laser beam within the depth of field. Assuming Lorenz-Mie's-theory for spherical water droplets allows for the classification into 30 size bins for particles within a defined sample area. The CAS sample area (SA) of $0.25 \pm 0.04$ mm$^2$ was determined by a water droplet beam mapping following the procedure outlined by Lance et al. (2010).  The size calibration has been performed according to Borrmann et al. (2000) and Rosenberg et al. (2012), leading to uncertainties of $\pm$ 16 % of the droplet diameters ($D_p$) reported here. The subsequent application of a Mie-binning proposed by Baumgardner et al. (2001) accounts for ambiguous assignments of scattering cross sections to corresponding droplet diameters leaving 18 unambiguous size bins. The time resolution of cloud measurements was set to 1 Hz. Coincident droplet measurements (Lance, 2012) were corrected with an empirical coincidence correction function derived by Kleine et al. (2018). The analyses of particle inter arrival times renders particle shattering (Field et al., 2003) negligible for this survey of warm low-level clouds. Cloud droplet number concentrations (CDNC) were derived from counted droplet number (N) within a given time period dt, size of the SA and aircraft's true airspeed (TAS):

$$CDNC = \frac{N}{SA \cdot TAS \cdot dt} \quad [cm^{-3}], \tag{1}$$

The cloud droplet effective diameter (ED) is then calculated as the ratio of the cumulated droplet radii as follows:

$$ED = 2 \cdot \frac{\sum_{i=1}^{n} N_i \cdot r_i^3}{\sum_{i=1}^{n} N_i \cdot r_i^2} \quad [\mu m]. \tag{2}$$

Finally, the liquid water content (LWC) is derived as the integral of the mass of spherical water droplets over all size bins, assuming a water density of one gram per cubic centimetre.

$$LWC = \sum_{i=1}^{n} CDNC_i \cdot 10^{-12} \cdot \frac{4\pi r_i^3}{3} \quad [g \; cm^{-3}] \tag{3}$$

A statistical comparison by Taylor et al. (2019) of cloud data from the cloud spectrometers flown on the 3-research aircraft during DACCIWA shows an excellent agreement in CDNC data from two cloud droplet probes (CDPs) and the CAS, with a mutual agreement of the medians and quartiles within 5 %. As for the ED, all three probes agreed within approximately 1 µm.

**Aerosol optical particle counter**

Aerosol number concentrations and size distributions in the size range of 0.25 µm up to 3 µm are measured with the optical particle counter SkyOPC (Version 1.129, GRIMM Aerosol, Germany) sampling inside the fuselage behind an isokinetic aerosol inlet. The instrument can detect particles starting from 0.25 µm in diameter, the upper size limit is set by particle transmission efficiency of inlet and tubing and was found to be between 1.5 µm and 3 µm depending on flight altitude (Fiebig, 2001, Schumann et al 2011). Water saturation dependent cloud condensation nuclei counters were not operated on the Falcon, therefore we use here the correlation between aerosol measured with the OPC and CO data as indicative of enhanced aerosol levels or pollution.

**Trace gas instruments for CO and H$_2$O**



Since data coverage of accumulation mode aerosol measurements with the SkyOPC is insufficient within clouds, CO concentrations have been used to derive location and dilution of urban emission plumes, factored by a vast amount of combustion products of organic matter within the plume (Haslett et al., 2019). CO mixing ratios were measured using infrared absorption spectrometry by the SPectrometre InfraRouge In-situ Toute altitude (SPIRIT, Catoire et al., 2017). The instrument

uses the effect of absorption in the mid-infrared spectrum by various trace gas species. The setup comprises three Quantum Cascade Lasers (QCLs) enclosed within a Robert cell, where two parabolic mirrors ensures a sufficient absorption path length. Sampled air is drawn from a rear facing inlet on the aircraft fuselage ducted via Teflon tubing to the instrument. Mass flow and system pressure are controlled by a scroll pump, an upstream regulator gage and a dosing valve. The absorption behaviour of an air sample is analysed with the help of a photodetector. The overall uncertainty for CO measurements is estimated to be

$\pm$ 4 ppbv, with a precision of $\pm$ 0.3 ppbv for flight measurements at a sampling time of 1.6 seconds (Catoire et al., 2017). Further trace gases to be measured with SPIRIT are $NO_2$ and $CH_4$ (Brocchi et al., 2019).

**Wind, temperature and humidity**

The Falcon basic measurement system captures the in-situ meteorological data (i.e. pressure, temperature, humidity and 3-dimensional wind) as well as the position and attitude information of the aircraft. The reliability of the pitot static system was

reviewed shortly before the DACCIWA campaign (Rotering, 2012). The measurement uncertainties for the pressure signals and subsequently the derived wind calculations are discussed in Bramberger et al (2017). They derive for the pressure signals a measurement uncertainty of 0.5 hPa, for the vertical wind 0.3 ms$^{-1}$ and about 0.9 ms$^{-1}$ for the horizontal wind components. The measurement uncertainty of the temperature signals is given with 0.5 K. However, due to wetting and evaporation biases the accuracy is significantly reduced within clouds. Details about the measurement techniques and the applied corrections are

described in Mallaun et al. (2015). Water vapour mixing ratios were measured with an uncertainty of $\pm$7 % with a CR-2 frost point hygrometer from Buck Research Instruments, LLC (Busen and Buck, 1995; Heller et al., 2017; Kaufmann et al., 2018) connected to a backward-facing inlet to exclude sampling of condensed water. The meteorological measurement system aboard the Falcon detected temperature and pressure with accuracies of $\pm$0.5 K and $\pm$0.5 hPa, respectively. From the water vapour and temperature data, the ambient relative humidity with respect to liquid (RHl) or ice (RHi) was calculated. In the air

surrounding the clouds RHl ranged near and below 100 % with an estimated uncertainty of $\pm$7 %.

**2.2 Meteorological data sets**

We use meteorological data analysis of the Integrated Forecasting System (IFS) of the European Centre for Medium-Range Weather Forecasts (ECMWF) Copernicus Atmosphere Monitoring Service (CAMS) to describe the basic meteorological conditions and atmospheric composition in the African tropics in summer 2016 for radiative transfer simulations.

The spatial resolution of the data set is ~ 80 km horizontally on 60 vertical levels from the surface up to 0.1 hPa. CAMS reanalyses are available on a 3- hourly time step. CAMS reanalysis was produced using 4DVar data assimilation in cycle 41r2 of ECMWF's Integrated Forecasting System (IFS) (Flemming et al., 2015). Here we use temperature, water vapor and other trace gas species as an input to the radiative transfer calculations.

**2.3 Radiative transfer calculations**

Radiative transfer calculations were carried out to show how the measured microphysical parameters of the boundary layer clouds affect the radiation budget of the atmosphere. Here, the radiative transfer solver DISORT (six-streams for irradiances) of the 1D routine UVSPEC from the program package *libRadtran* (Mayer and Kylling, 2005) is used to calculate the effects of polluted and less polluted water cloud layers in the planetary boundary layer on static vertical profiles of irradiances and heating rates in the shortwave (SW from 0.24 to 5.0 µm) and the longwave spectral range (LW from 2.5 µm to 100 µm).

Considered are all relevant radiative processes in the atmosphere-surface system, i.e. multiple scattering, absorption, and



thermal emission by atmospheric aerosol particles, cloud droplets, ice crystals, trace gases, and the ground. The microphysical properties of cloud layers defined in *libRadtran* are described by the vertical profiles of the ED and LWC based on the observations. Below cloud base and above cloud top these values are set to zero. For the low-level water cloud, radiative transfer calculations are based on the vertical statistics on the LWC and ED profiles representing the cloud fields probed during the Falcon operation days. The low-level cloud layer is embedded between 790 m and 1870 m and discretised into 60 model layers. To show the pure effect of different ED profiles on the radiation fields it is assumed that the LWC profile for the polluted low-level water cloud is the same as for the less polluted cloud (see Sect. 4).

To study the effect of cloud layers overlying the boundary layer cloud, a thin medium-high water cloud and an ice cloud are added to the model atmosphere in a sensitivity study. The medium-high cloud is located between 4770 m and 4840 m, the ice cloud between 9100 m and 9700 m, as found on average in the measurements. LWC/IWC and ED profiles for these clouds stem from single measurements sometime during the campaign and are then gradually adjusted for sensitivity studies (section 5.2). The translation of IWC and ED to the optical properties and the description of the ice crystals as a general habit mixture follow Baum et al. (2005a). In order to pinpoint the sole contribution to a cloud radiative effect only one representative vertical profile as on 29 June, 2016 for the meteorological parameter temperature and pressure, and such air density, and the profiles of the trace gases $H_2O$, $O_2$, CO, $CO_2$, and $NO_2$ are taken from ECMWF CAMS data (Fleming et al., 2015). For $CH_4$ and $N_2O$ the default mixing ratios of *libRadtran* are chosen, i.e. 1.6 ppmv and 0.28 ppmv, respectively. The 12:00 and 18:00 UTC CAMS analysis are interpolated in space and time to the flight track of the Falcon on 29 June. Selected are vertical profiles close to the location of Lomé (Togo). The temperature profile used was compiled as a composite from all campaign flights. Regarding $O_3$, GEMS data are used which include background ozone chemistry. All profiles are assumed to be constant over time.

To describe the reflection properties of the surface in the SW spectral range the MODerate-resolution Imaging Spectroradiometer (MODIS) broadband albedo product (MCD43C3) has been used (Schaaf, and Wang, 2015). To be close to the cloud conditions during the campaign the bihemispherical reflectance (white-sky albedo) has been selected. The MODIS product contains 16 days of data provided in a 0.05-degree (5600 m) latitude/longitude grid. Selected MODIS albedo data represent the 16 days from 29.06. to 15.07.2016, thus covering the campaign period very well. Spatial averaging of the broadband white-sky albedo data over an area including the flights tracks gives α = 0.16. In the LW range a constant emissivity of ε = 0.99 is assumed which after Wilber et al. (1999) should represent surfaces consisting of types like savanna and urban quite well. The aerosol optical thickness (AOT) has been derived from Aeronet data for July 2016 based on measurements at the sites Koforidua_Anuc (Ghana) and KITcube_Save (Benin). Averaging the Aeronet data results in AOT = 0.38 at 0.55 µm. For the atmospheric boundary layer, the *libRadtran* aerosol type "urban" has been selected due to increased emissions from industry and the transport sector in the region around the main cities.

### 2.3.1 Radiative quantities

One basic model output is spectral irradiance $F_{\Delta\lambda}$ in $Wm^{-2}$ at each layer boundary from which the net irradiance $F_{net,\Delta\lambda}$ is calculated according to:

$$F_{net,\Delta\lambda} = F_{down,\Delta\lambda} + F_{up,\Delta\lambda} \quad [Wm^{-2}] \tag{4}$$

$F_{up,\Delta\lambda}$ is the upward and $F_{down,\Delta\lambda}$ the downward directed irradiance integrated over the wavelength interval $\Delta\lambda$ (here: SW, LW). Upward directed irradiances are counted negatively, downward directed irradiances positively. The net radiative forcing $RF_{net}$ in $W/m^2$, as used in this study, is defined as the difference of the net irradiances calculated for the atmosphere with an embedded polluted cloud (CO >155 ppbv) minus the net irradiances obtained for the less polluted cloud. Net irradiances are balanced over the SW and LW spectral ranges:

$$RF_{net} = \left[F_{net,SW,CO>155ppbv} + F_{net,LW,CO>155ppbv}\right] - \left[F_{net,SW,CO<135ppbv} + F_{net,LW,CO<135ppbv}\right] \quad [Wm^{-2}] \tag{5}$$



The ratio of the upward directed irradiance to the downward irradiance at each model layer is defined as the albedo integrated over the wavelength interval $\Delta\lambda$:

$$\alpha_\lambda = \frac{F_{up,\Delta\lambda}}{F_{down,\Delta\lambda}} \tag{6}$$

The heating rate $H_{\Delta\lambda}$ in K/day is integrated over the spectral interval $\Delta\lambda$ (here: SW, LW) and describes the temperature change of a model layer within a given time interval. It is defined as

$$H_{\Delta\lambda} = -\frac{1}{\rho_{air} \, c_p} \frac{\vartheta F_{\Delta\lambda}}{\vartheta z} \; [K \, day^{-1}] \tag{7}$$

with $\rho_{air}$ denoting the air density, $c_p$ the specific heat capacity of air at constant pressure. $\vartheta \, F_{\Delta\lambda} \, / \, \vartheta \, z$ represents the vertical divergence of the net irradiance $F_{net,\Delta\lambda}$. Summing up $H_{\Delta\lambda}$ over the SW and LW spectral range results in the net heating rate
$H_{net}$.

**3 Microphysical properties of tropical continental low-level clouds**

Cloud measurements were performed with the CAS instrument during the 12 research flights led from Lomé, Togo across the West African countries of Côte d'Ivoire, Ghana and Benin, with flight segments of above the Gulf of Guinea (Figure 2).  In order to characterize city plumes of the metropolitan regions Lomé (Togo), Accra (Ghana), Abidjan (Ivory Coast) Save
(Benin), sampling was performed during circuit flight patterns, covering the inflow as well as the outflow region of a city. The probing of the Lomé city plume was emphasised with multiple recurrent flights. Measured CDNC along the flight tracks (Figure 2) in the Lomé downwind region to the east of the city uncover enhanced CDNC above 700 cm$^{-3}$. Similar observations have been made around other close by coastal cities in the region, with small CDNC below 70 cm$^{-3}$ over coastal areas and enhanced CDNC further inland with concentrations between 210 cm$^{-3}$ and 490 cm$^{-3}$ and enhanced CDNC in the vicinity or
downwind region of conurbations. The density distribution of measured CDNC in low level clouds, i.e. between ground and 1800 m (Fig. 3) peaks between 100 cm$^{-3}$ and 150 cm$^{-3}$ with a median value around 250 cm$^{-3}$. Towards the upper end of the distribution CDNCs of 900 cm$^{-3}$ up to 950 cm$^{-3}$ with a frequency of 1% of occurrence can be found. Above that CDNCs around 1300 cm$^{-3}$ account only for less than 0.2 %.

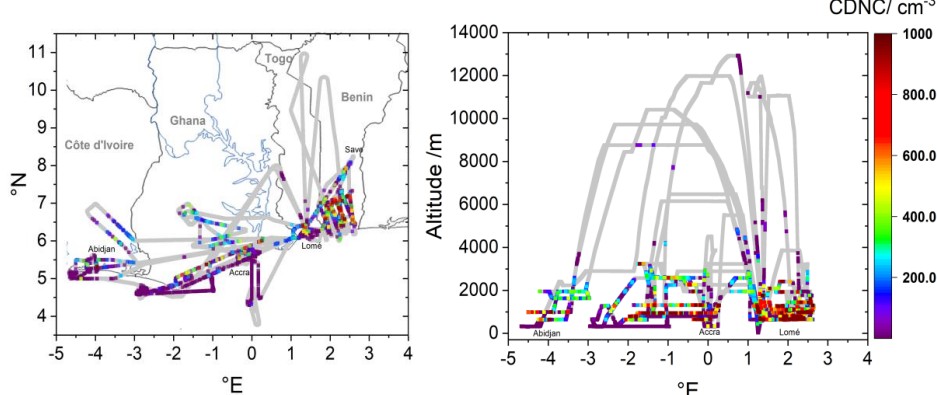

**Figure 2: Flight paths of the DLR Falcon research aircraft during the DACCIWA airborne measuring campaign from 29 June until 14 July 2016. Based in Lomé, Togo 12 research flights in total led across the West African neighbouring countries of Côte d'Ivoire, Ghana and Benin, as well as segments of above the Gulf of Guinea.  Measured cloud sequences from the CAS-DPOL are colour-**
**coded along the flight tracks.**





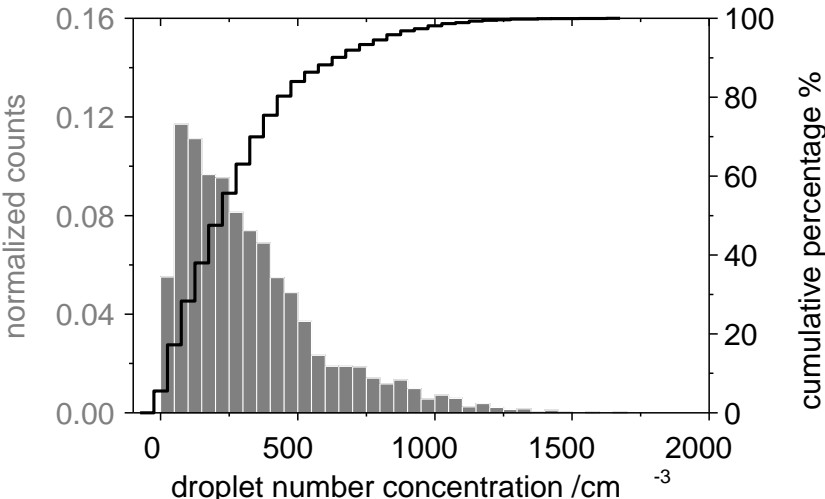

**Figure 3: Density distribution of measured cloud droplet number concentrations in low-level clouds across West Africa during the DACCIWA campaign as normalised counts and cumulative percentage.**

### 3.1 Cloud classification for polluted and less polluted clouds

Measurements of cloud condensation nuclei (CCN) are unreliable within clouds as activated CCN have grown to sizes too large to enter the aerosol inlet line and are therefore not measured by the OPC. In addition, the OPC lower cut-off size of 250 nm just measures the larger CCN. Therefore, carbon monoxide is used as a pollution marker and as a proxy for the amount of accumulation mode aerosol in order to distinguish polluted from pristine or at least less polluted low-level clouds. Figure 4 shows a linear correlation between CO and accumulation mode aerosol number concentrations outside clouds between 700 m

and 1800 m altitude. Although the lower part of the accumulation mode aerosol might not lie within the measuring range of the SkyOPC-Probe, both parameters

correlate linearly. This motivates our assumption that high CO mixing ratios can be regarded as a valid proxy for the abundance and number concentration of accumulation mode aerosol, acting as potential CCN.  Though, CO is a long-lived species, that does not react with the cloud drops, accumulation mode aerosols as cloud condensation nuclei might agglomerate and sediment

or are subject to wet deposition and undergo a change in chemical composition (Capes et al. 2008). Also, different aerosol sources might affect the relationship between CO and aerosol measured with the OPC. Deviations from the overall linear correlation could be attributed to these effects. Throughout the campaign almost 40 % of the measured CO concentrations lie between 140 ppbv and 150 ppbv. The mean value of all CO mixing ratio measurements below 1800 m is 151 ppbv, with a median of 143 ppbv. This CO mixing ratio can be regarded as a CO seasonal average, originating from either




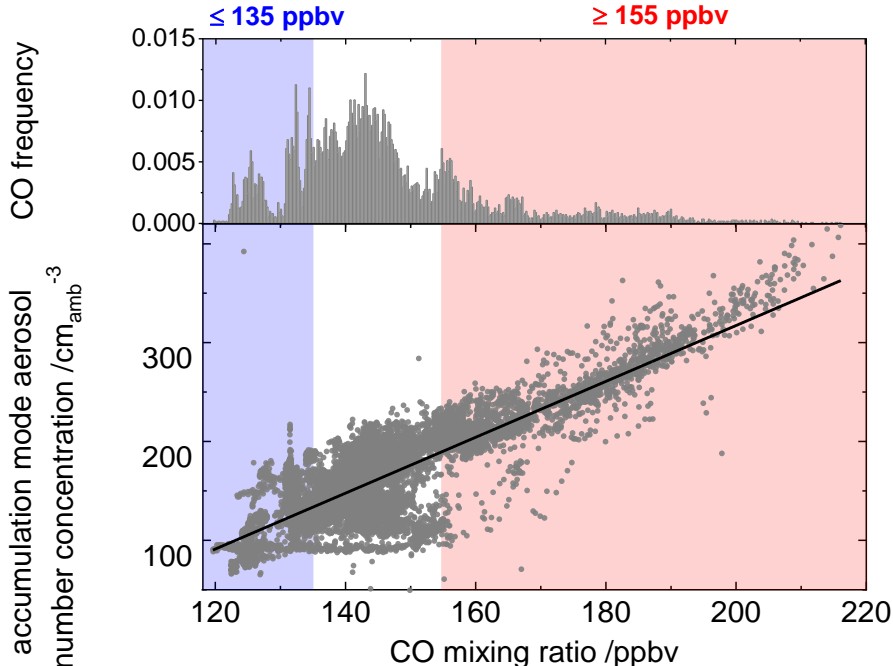

**Figure 4:** Correlation of accumulation mode aerosol and CO mixing ratio from all flights of the DLR Falcon with a linear best fit of $N_{acc\text{-}aerosol}$= -248.05 ± 2.65 + 2.83 ± 0.02 · CO [ppbv]. The CO mixing ratio hence is used as a proxy for the degree of pollution and the abundance of activated cloud condensation nuclei within low-level clouds. Henceforth clouds with CO levels ≤135 ppbv (22nd percentile) and ≥155 ppbv (79th percentile) are characterised as less polluted or substantially polluted, respectively.

local, as well as remote sources in southern Africa (Haslett et al., 2019). In order to distinguish between polluted and less polluted air masses, we defined a threshold with CO mixing ratios lower than 135 ppbv (22nd percentile) as moderately polluted air masses and CO mixing ratios larger than 155 ppbv (79th percentile) as substantially polluted air masses. This nomenclature follows a classification used by Wendisch et al. (2016) from the ACRIDICON campaign in Brazil. In light of median CO mixing ratios of 75 ppbv above the South Atlantic Ocean, as measured on the CLARIFY-2017 campaign (Haywood et al., 2021), the measured CO average mixing ratio during DACCIWA has to be regarded as moderately polluted. The correlation between CO and accumulation mode aerosols below 1800 m and outside of clouds are analysed in Figure 4. CO measurements range between 120 ppbv and 300 ppbv. We find a linear correlation between aerosol number concentrations and CO with a slope of 2.83 (± 0.02) and a y-intercept of -248.05 (± 2.65) with a $R^2$ of 0.83.

## 3.2 Microphysical properties of tropical continental low-level clouds and influence of pollution

Particle size distributions of low-level clouds are shown in Figure 5 for all cloud data measured in the boundary layer below 1800 m altitude. The cloud data were grouped into less polluted background clouds and polluted clouds using CO as pollution tracer. As already shown by Taylor et al. (2019) and Haslett et al. (2019), the less polluted clouds are still strongly influenced by biomass burning aerosol from southern Africa. Despite the high background CO mixing ratio also for the less polluted clouds, a difference is found in the cloud droplet size distribution of both cloud classes.

The mean effective diameter of the noon-time low-level less polluted clouds is 14.8 µm. The polluted clouds have a smaller ED of 12.4 µm, higher pollution levels tend to lead to an increase in cloud drop number and a decrease in cloud particle size, in our case a 17% reduction in particles size has been detected. The maximum of the particle size distribution of the polluted



cloud is shifted towards smaller diameter, compared to the less polluted case. Variations between the size distribution decrease towards larger droplet diameters. A more accurate determination of the mode maximum in the substantially polluted case is limited due to Mie ambiguities in the lower size range. The majority of substantially polluted clouds were encountered in the city outflow regions along the West African coast. Variations in updraft speed have been shown to have a significant impact

on cloud drop number concentrations, in particular at low updraft speeds, e.g. Moore et al., 2013, Taylor et al., 2019; Braga et al., 2017a, b, 2022; Cecchini et al., 2017; Dadashazar et al., 2021, Kirschler et al., 2022). In order to investigate potential effects of updraft speed variations on cloud droplet number concentrations, we derived probability density functions of the updraft speeds measured with the Falcon basic measurement system for the polluted and less polluted cloud cases.

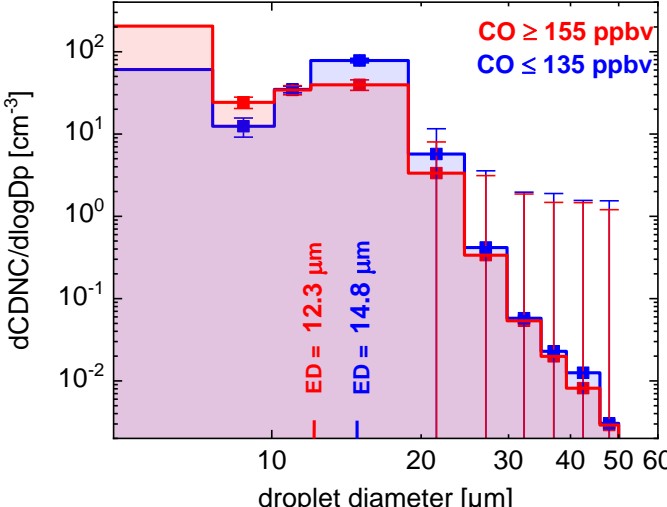

**Figure 5: Average cloud droplet size distributions of low-level clouds below 1800 m over West Africa from the Cloud and Aerosol Spectrometer (CAS) measured during the DACCIWA campaign in summer 2016, separated into less-polluted (blue) and substantially-polluted (red) clouds according to CO mixing ratios ≤135 ppbv and ≥155 ppbv, respectively. The less-polluted clouds have an effective cloud droplet diameter (ED) of 14.8 µm, the substantially-polluted clouds have an ED of 12.3 µm.**




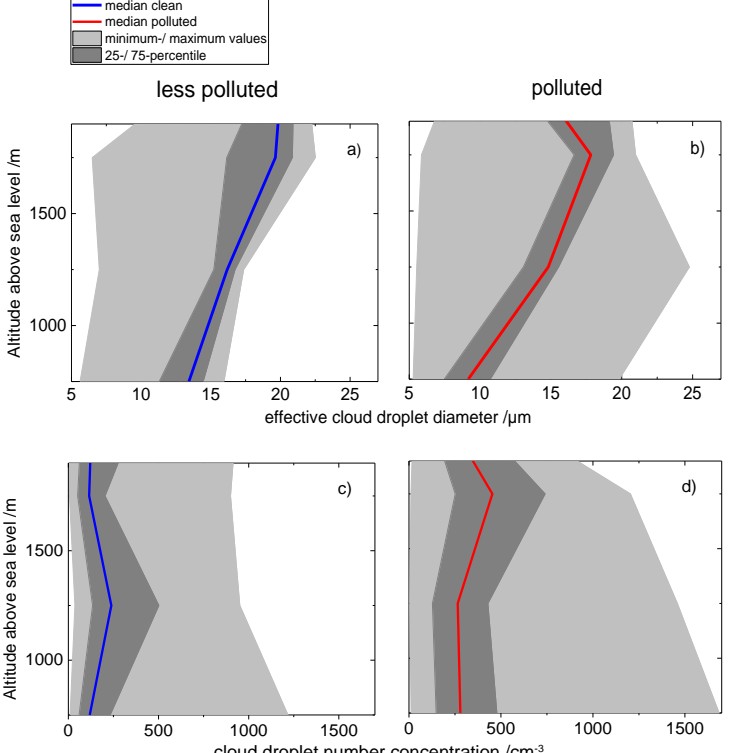


**Figure 6: Vertical statistics of cloud microphysical properties for less polluted and polluted clouds: (a and b) cloud effective diameter and (c and d) cloud droplet number concentration from measurements of low-level clouds at various heights and various cloud bases.**

The mean updraft of 22 cm/s indicates low updraft velocities in the noontime within low-level clouds. There is no statistically significant difference in updraft speed distributions for the high and less polluted clouds. For our study, differences in updraft

speed distributions are small for the polluted and less polluted case and have a minor influence on changes in cloud particle size distributions.

## 4 Altitude and Aerosol effect on low-level clouds

An analysis of a vertical dependency on microphysical properties to reveal relevant cloud features in low-level clouds is shown in Figure 6. The profiles of CDNC and ED of the less polluted (CO ≤135 ppbv) and the polluted clouds (CO ≥155 ppbv),

compiled as averages of all low-level cloud measurements in various heights and with various cloud bases, show smaller cloud droplets in the polluted clouds over the complete altitude range below 1800 m and generally higher CDNC. The less polluted clouds already show larger ED in the lowermost cloud layer. The median ED as generally dependent on height above cloud base increases almost steadily up to 19 μm. The largest difference in ED between both cloud classes is visible in the lowermost layer, with median values of 9 and 14 μm. The polluted case also reaches its largest droplet radius at 1700 m altitude, but

shows smaller droplet diameters of 16 μm compared to 19 μm for the less polluted case.

Generally lower droplet number concentrations were observed for the less polluted clouds compared to the substantially polluted clouds. The less polluted clouds have a median CDNC of 110 cm$^{-3}$ in the lowest cloud layer and the median CDNC





increases to near 200 cm$^{-3}$ in the middle of the cloud. A decrease in CDNC is measured at the top of the cloud layer. In the polluted case, near surface values of CDNC are already increased compared to the less polluted clouds. Starting from 250 cm$^{-3}$ (median) in the lowest altitudes, CDNC show a slight decrease towards the mid of the cloud and a distinct increase of the droplet number concentration in the higher cloud layer up to 440 cm$^{-3}$. Averaged over the entirety of low-level clouds in the less polluted case, median CDNC of 240 cm$^{-3}$ were measured, with a variability as indicated by the 25 and 75 percentiles of 52 cm$^{-3}$ to 501 cm$^{-3}$. The polluted clouds have higher mean CDNC of 324 cm$^{-3}$ and a 25 to 75 % range of 60 cm$^{-3}$ to 740 cm$^{-3}$. Hence, the polluted clouds contained 35% more cloud drops than the less polluted reference case. An aerosol effect on clouds and its implications for the radiation budget (Twomey, 1991) could lead to an increase in cloud droplet number concentration, at the same time reducing the effective radius, given that the liquid water content is similar (since LWC is basically given by the amount of condensable water and thus by the updraft speed).

**5 Net radiative forcings and heating rates induced by tropical low-level clouds**

Figure 7 presents the diurnal cycle of the net radiative forcing, $RF_{net}$, at the top of the atmosphere (TOA) as the difference of the net irradiances between the polluted and less polluted case for the horizontally homogeneous water cloud in the planetary boundary layer. The cycle of the solar zenith angle (SZA) corresponds to day 180 (28 June 2016) as of the beginning of the flight campaign. Shown are the results from 06:00 to 18:00 UTC. During night the net forcing is determined solely by the irradiances in the longwave spectral range leading to a temporally constant value of $RF_{net} = +1.1$ W m$^{-2}$ according to the assumption of a fixed temperature profile. During daytime the net forcing depends on the SZA and is negative with a minimum of $RF_{net} = -16.3$ W m$^{-2}$ at 12:00 UTC. The SW forcing dominates the LW forcing during daytime. Negative forcing values are caused by a higher cloud albedo in the polluted case, a consequence of smaller effective diameter. At 12:00 UTC (08:00 UTC) the SW albedo (Equ. 6) at the top of the cloud (TOC) is 0.81 (0.85) for the polluted cloud against 0.79 (0.84) for the less polluted cloud.

Averaging the net forcing calculated for TOA over 24 hours gives $RF_{net} = -3.9$ W m$^{-2}$. Note, all numbers for $RF_{net}$ given in this section are valid for a horizontally homogenous cloud layer of 100 % coverage. The net forcing, which, as defined in this study, is based on the pollution effect only, decreases with decreasing cloud cover.

As Hill at al. (2018) have shown that low-level clouds often coincide with medium- and high-level cloud layers, we perform

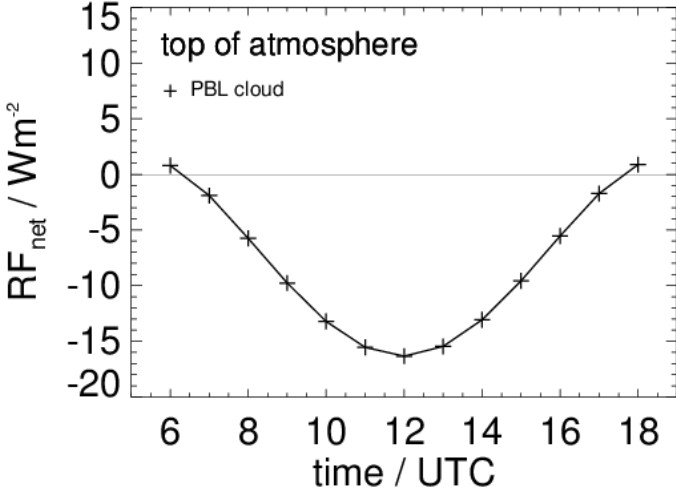

**Figure 7: Diurnal cycle of net radiative forcing ($RF_{net}$) at the top of the atmosphere for 29 June 2016 containing one horizontally homogeneous water cloud in the planetary boundary layer (PBL cloud) between 790 and 1870 m.**





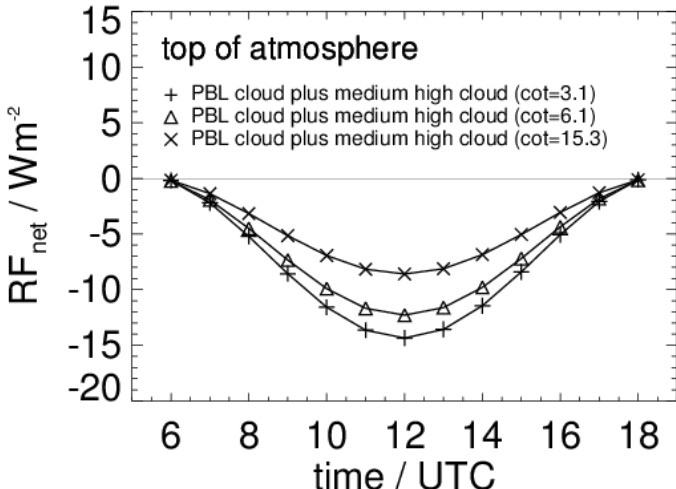

**Figure 8: As in Figure 7, but for the atmosphere containing two water cloud layers, i.e. one cloud in the planetary boundary layer (PBL cloud) located between 790 and 1870 m and medium high clouds of different optical thicknesses located between 4770 – 4840 m. The curve marked with crosses results for the medium high cloud of measured optical thickness COT = 3.1 at 0.55 μm, the two other curves represent net radiative forcings for artificially increased COT values as indicated.**

a sensitivity study on the net $RF_{net}$ including clouds at higher altitudes above a low-level cloud layer with polluted and less polluted microphysical properties. Figure 8 shows diurnal cycles of the TOA net forcing for an atmosphere containing a second, geometrically thin water cloud located between 4770 m and 4840 m above the boundary layer cloud. The microphysical properties of this cloud stem from a single measurement and pollution effects for this cloud are not considered. The LWC and ED profiles result in a cloud optical thickness of COT = 3.1. Figure 8 shows that scattering processes at this second medium-high cloud layer slightly dampen the TOA net forcing caused by the pollution of the boundary layer cloud. At 12:00 the model gives $RF_{net}$ = -14.4 W m$^{-2}$ compared to $RF_{net}$ = -16.3 W m$^{-2}$ for the atmosphere containing only one cloud layer (Fig. 7). Increasing the optical thickness of the medium-high cloud to COT = 6.1 (COT = 15.3) by increasing the LWC by a factor of 2 (5) in each layer, in order to estimate the sensitivity of this additional cloud layer regarding the net radiative forcing, results in $RF_{net}$ = -12.3 W m$^{-2}$ ($RF_{net}$ = -8.6 W m$^{-2}$) at 12:00 (Fig. 8). Averaging over 24 h reduces TOA net forcing values to -4.0 W m$^{-2}$ (COT = 3.1), -3.4 W m$^{-2}$ (COT= 6.1), and -2.4 W m$^{-2}$ (COT = 15.3). A smaller coverage of the medium-level cloud over the homogeneous boundary layer cloud has less impact on the net forcing; conversely, a homogeneous medium-level cloud over broken boundary layer clouds will reduce the net forcing values at TOA.

Finally, simulations were carried out for an atmosphere with an additional ice cloud layer located between 9100 m and 9700 m. Since the measured microphysical parameters lead to an optical thickness clearly below 0.1, COT has been increased to 0.3 by manipulation of the IWC while keeping the vertical extension of the ice cloud. As expected, the additional effect of an optically thin ice cloud layer on the TOA net forcing turns out to be weak. For example, simulations that take into account three cloud layers give $RF_{net}$ = -12.0 W m$^{-2}$ at 12:00 in comparison to $RF_{net}$ = -12.3 W m$^{-2}$ valid for the two-cloud layer configuration with a medium-high cloud of COT = 6.1.

In order to get an idea how both, polluted and less polluted, low-level clouds independently locally affect a surrounding temperature profile, Figures 9 a and b display diurnal variations of instantaneous net heating rates $H_{net}$ at the top of the boundary layer cloud (TOC) and at the surface, respectively. At TOC (Fig. 9a) and at the surface (Fig. 9b) the LW cooling dominates the SW heating during daytime, de facto leading to net cooling rates at both altitudes. At 12:00 UTC the polluted cloud (less polluted cloud) gives a LW cooling rate of $H_{LW}$ = -342.4 K day$^{-1}$ (-317.8 K day$^{-1}$) at the TOC whereas the corresponding SW





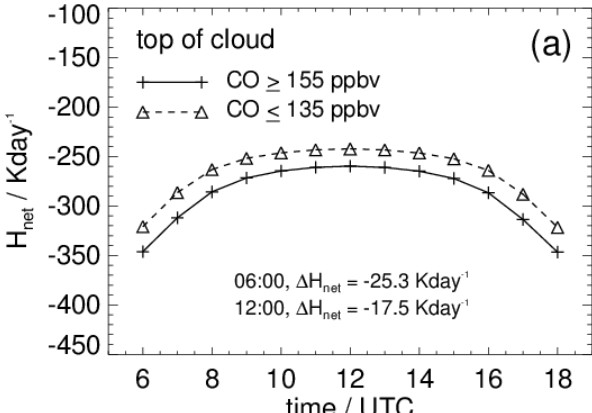

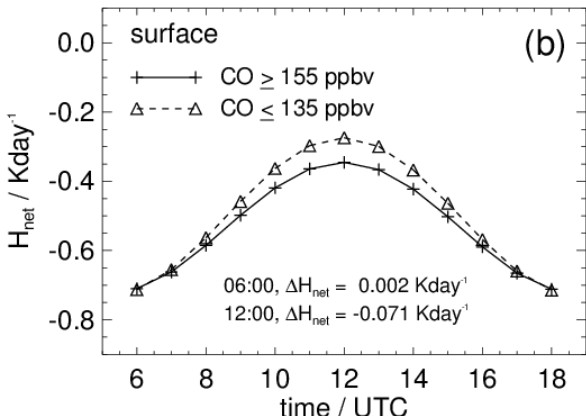

**Figure 9: Diurnal cycles of instantaneous net heating rates ($H_{net}$) at the top of the boundary layer cloud (a) and at the surface (b) for the polluted (CO ≥155 ppbv) and the less polluted atmosphere (CO ≤135 ppbv). $\Delta H_{net}$ denotes the difference of $H_{net}$ for the polluted and the less polluted case at the times indicated.**

warming rate results in $H_{SW}$ = +82.7 K day$^{-1}$ (+75.6 K day$^{-1}$). With increasing SZA the SW warming decreases converging to the value of the LW cooling rate. The curve for the polluted atmosphere shows higher cooling rates compared to that of the less polluted atmosphere (Fig. 9). Reason is the higher optical thickness of the polluted model cloud (COT = 85.7 vs. COT = 75.6) leading to a greater vertical divergence of net irradiances (Equ. 7) at the top of the cloud. Differences in net heating rates between both cases range between -17.5 K day$^{-1}$ at noon and -25.3 K day$^{-1}$ at 6 UTC. In case of an atmosphere containing the low and the medium-level water cloud the cooling rate difference results in 0.1 K day$^{-1}$ at the top of the medium-high cloud at 12:00 UTC (not shown). At the surface (Fig. 9b) cooling rates are strongly reduced due to the overlying homogenous cloud which causes very small divergences of net irradiances. Cooling rate differences due to pollution effects are almost negligible; the maximum difference at 12:00 is -0.07 K day$^{-1}$.

**6 Discussion**

This study presents a comprehensive data set of in-situ cloud measurements from the research aircraft Falcon 20 in tropical West Africa during the monsoon onset season. This data set contributes to fill the gap of scarce measurement coverage in this region at the Gulf of Guinea, ranging as far as Benin to the east and Côte d'Ivoire in the west. The characterisation of low-level clouds (smaller 1800 m altitude) across this region, between 29 June and 14 July 2016 shows median cloud droplet





number concentrations around 270 cm$^{-3}$, as measured from the CAS underwing probe. To identify aerosol-cloud effects we
aimed to classify the probes air according to an apparent pollution level in terms of accumulation mode aerosol number
concentration and relate these directly to respective clouds. Because measurements within clouds are flawed once CCN are
activated, we correlated aerosol number concentrations (>0.25 µm) outside clouds with CO mixing ratios measured with the
SPIRIT absorption spectrometer aboard the Falcon. A linear trend shows the relation between these two parameters. Using the
CO mixing ratio as a proxy for the pollution level of an air mass, we regard CO ≤135 ppbv and CO ≥155 ppbv, as less polluted
and substantially polluted, respectively. Although a high biomass burning aerosol background entrained from Central Africa
reduces the delta between both classes, an aerosol-cloud effect is still visible in cloud microphysical properties, showing
effective droplet diameter between 12.3 µm in the polluted case, compared to an ED of 14.8 µm in the less polluted case and
CDNCs that almost double (median) in the polluted case.

The influence of the pollution level on the local radiative budget as a net forcing and derived instantaneous heating rates
between both cases of low-level clouds, was subject of a modelling study with the *libRadtran* 1D-radiative transport model
under application of the DISORT solver. Seeding the simulation with the microphysical properties from the in-situ
measurements show a negative net forcing at TOA (CO ≥155 ppbv minus CO ≤135 ppbv) for the entire day for a low-level
boundary layer cloud. Thus, in a polluted atmosphere changes due to ED would locally lead to a further cooling of the earth-
atmosphere system, with a net forcing of -16.3 W m$^{-2}$ at noon and an 24 h average of -3.9 W m$^{-2}$. To put this into perspective:
A comparison in net radiative forcings at TOA of a substantially polluted cloud layer with a cloud free scenario, as reference
baseline, results in -638.2 W m$^{-2}$ at 12:00 UTC with a 24 h average of -196.6 W m$^{-2}$, the LW contribution is +9.4 W m$^{-2}$. Thus,
net forcing at TOA (CO ≥155 ppbv minus CO ≤135 ppbv) accounts for only 2.6 % of the net forcing of the reference case
(CO ≥155 ppbv minus cloud-free). This small difference in net radiative forcing caused by different cloud microphysics shows
the non-linear relationship of the processes between radiative balance of differently polluted low-level clouds and resulting
radiative forcings, especially when considering an apriori polluted ground state.

As the occurrence of layered cloud structures with medium and high-level clouds atop a low-level cloud layer still poses an
uncertainty in weather models, this scenario was also simulated, suggesting that net forcing effects due to pollution
(CO ≥155 ppbv against CO ≤135 ppbv) are discernible at TOA. Maximum net forcing values at TOA range between -
16.3 W m$^{-2}$ (one cloud layer) and -8.5 W m$^{-2}$ (three cloud layers), with a 24 h-average of these maximum forcings at TOA
divided by a factor of about 3.5. The direct effect of this net radiative forcing translated into an instantaneous net heating rate
(SW+LW) at TOC of a low-level less polluted compared to a substantially polluted boundary layer cloud changes by about -
18 K day$^{-1}$ at 12:00 UTC and -25 K day$^{-1}$ at 06:00 UTC. Again, the net heating rate of the scenario with vertically layered
clouds with each cloud sub set as low-level cloud show that changes due to pollution are negligible at the surface and at TOC.
These results confirm observations by previous studies and not only extend them by a detailed analysis of radiative effects of
low-level clouds under consideration of different pollution levels, but also the presence of layered cloud structures with a
variation of the pollution level within the lower most cloud layer. This study confirms that the climate sensitivity towards a
projected steady increase in aerosol emissions and such cloud condensation nuclei concentrations in the boundary layer, due
to proceeding urbanisation and growth of the West African economies might be damped, as the background aerosol loading,
at least at the time of measurement, is already significantly enhanced.

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



**Author contributions:** V.H, J.K., D.S., S.K., M.M. performed the measurements and analysed the data. V.H. and C.V. performed the scientific study, R.M. conducted radiative transfer calculations and V.H. and R.M. wrote the paper. All authors contributed to the manuscript.

**Competing interests:** The authors declare that they have no conflict of interest.

**Acknowledgements:** We thank the DLR flight department including the sensor and data team for excellent support and flight operations in West Africa during DACCIWA.

**Funding**

EU DACCIWA funding information. CV and MM are funded by the Deutsche Forschungsgemeinschaft (DFG, German Research Foundation) under SPP PROM Vo1504/5-1 and SPP HALO1294 1504/7-1 and under TRR 301 – Project-ID 428312742.