# Peer review of "Pollution slightly enhances atmospheric cooling by low-level clouds in tropical West Africa"

_Atmospheric Chemistry and Physics, 2022_

## Author Comment (AC1)

**"Pollution slightly enhances atmospheric cooling by lowlevel clouds in tropical West Africa", submitted to ACP by Valerian Hahn et al., 2022**

We thank Referee #1 for his valuable comments and the effort that has been put in reviewing the submitted manuscript on the pollution effects and atmospheric cooling by low-level clouds in tropical West Africa.

**General Comment 1:**

A. Introduction: I don't understand the primary motivation for this study. There are lots of nice details about prior work but no clear story about what it all adds up to and where important knowledge gaps remain. Why is this new work needed? How does it relate to the previous work?

**Answer General Comment 1:**

We thank Referee#1 for the indication to emphasize the motivation of this study. To get a better understand of the basic idea for the analysis of the microphysics of low-level clouds in West Africa, one has to see the underlying scientific questions tied to the design of the DACCIWA campaign. As Knippertz et al. (2015) outline tropical West Africa faces a major population growth and urbanization. The effect of an increase in anthropogenic emissions is not only of concern in view of health aspects, but poses an uncertainty for future local climate. Knippertz et al. (2015) suggest that clouds above West Africa could be 'highly susceptible to increases in anthropogenic pollution'.

Previous studies found discrepancies in local weather models, also associated with a misrepresentation of multilayered cloud structures in this region.

The contribution of low-level clouds to the energy budget during the monsoon onset phase in West Africa was a significant scientific question to be answered by the DACCIWA campaign. This is where a knowledge gap was identified by the DACCIWA consortium und our study aims at contributing to fill this gap.

Our study can be very well embedded in the body of existing studies and expands their current state of knowledge within the scope of the DACCIWA campaign, such as Haslett et al. (2019), who analyzed the aerosol provenance, van der Linden et al. (2015) and Hill et al. (2018), who investigated the occurrence of cloud structures and patterns from remote sensing retrievals and latter calculated an overall radiation budget. Whereas Taylor et al. (2019) focus on a statistical analysis of in-situ cloud data and distinguish between local and continental sources (inland) as well as transported maritime air masses, our study makes a distinction between polluted and less polluted clouds, based on CO mixing ratios, which we can reference to accumulation mode aerosol. Furthermore, Taylor et al. (2019) explained with a sensitivity study using a parcel model the observed difference between maritime and inland low-level clouds.

What is left unanswered to this point is the role of polluted and less polluted low-level clouds within the regional radiation budget. Our study extends the current body of literature by adding radiative transfer calculations fed with a derived representation of measured low-level clouds under consideration of the degree of pollution involved in air associated with the genesis of and/or entrainment in low-level clouds. We look at how the Twomey effect influences the instantaneous radiative forcing and instantaneous heating rates. Although we cannot simulate and evaluate overall cloud adjustments with our RT model, in order to come up with an effective radiative forcing, this still is a crucial step towards classifying the microphysical and radiative influence of increased aerosol emissions on low-level clouds during the monsoon onset period in tropical West Africa.

A greater emphasize on the significance of our study and its aim to answer a major scientific question of the DACCIWA campaign will be included in the manuscript.

**General Comment 2:**

B. Cloud adjustments and radiative forcing: It would be helpful to be more precise in the discussion here. What you are calculating is the instantaneous radiative forcing due to the Twomey effect alone (effect of greater aerosol leading to greater cloud droplet number concentration and smaller effective radius for the same amount of cloud water). It neglects both cloud adjustments to the Twomey effect and other atmospheric adjustments to the changes in heating profiles and thus is not the effective radiative forcing, which is a distinction worth pointing out. It also may be worth thinking about the effect of aerosols on the clear-sky fluxes, as the "clean" and "polluted" cases would presumably also have different AOD values.

**Answer General Comment 2:**

The distinction between an instantaneous radiative forcing and the effective radiative forcing as Referee#1 emphasizes is an important one to be made. Surely, this point should be underlined when discussing the RT model results. We will review the manuscript in order to make sure to use the terms instantaneous net radiative forcing and instantaneous heating rates throughout the entire manuscript correctly. The investigation of cloud and atmospheric adjustments according to the demonstrated aerosol-cloud interaction cannot be simulated in the discrete RT-model libRadtran and lies outside the scope of our study.

Analyses of adjustment mechanisms with data from the DACCIWA period were analyzed e.g. by Pante et al. (2021), who find a likely cause of reduced precipitation caused by enhanced aerosol levels, but a cloud lifetime effect, such as shown by Christensen et al. (2020) has not been analyzed. Nevertheless, cloud and ongoing atmospheric adjustments are already implicitly included in the measured data set. Just like Douglas and L'Ecuyer (2020) a determination and quantification of cloud adjustment processes need a more holistic approach and might feature a synopsis of additional data sources and measurements to in-situ data alone, such as remote sensing and microphysical models.

The question of a varying AOD is interesting in the context of determining effective radiative forcing when quantifying of the influence of local sources versus aerosol sources from long-range transport. However, our study specifically examines the influence of a ubiquitous aerosol background and its influence on cloud microphysics and the mere aerosol-cloud effect on instantaneous radiative forcing.

This study investigates a significant open research question posed by the DACCIWA consortium prior to the campaign, and extends other studies by performing radiative transfer calculations.

Our findings provide an excellent starting point for further model studies, e.g. to determine an effective radiative forcing of the overall atmospheric system in the region.

**General Comment 3:**

C. Cloud specification in RT model: In general, does the ~1 km cloud make sense? Is that how thick the clouds typically are? It would be helpful to perhaps discuss distributions of cloud top and base heights for the polluted and clean cases. This would also be relevant for thinking about how to interpret the vertical profiles in Figure 6 (see comment below).

**Answer General Comment 3:**

We thank the Referee for the suggestion to include a discussion of the distribution of cloud top and cloud base heights in the manuscript.

The determination of the vertical cloud structure is a representation of a height averaged profile over all campaign flights with the DLR-Falcon, including cloud base height (CBH) at an average of 666.29 ±82.82 m and cloud top height (CTH) around 1909.40 ±440.12 m. While interpreting the vertical profile, one has to consider the underlying measuring and flight principles inherent in aircraft-based in-situ measurements: Contrary to genuine vertical profiles e.g. from balloon soundings, aircraft measurements incorporate a much larger horizontal component. An aircraft based vertical atmospheric survey only features measurements from altitudes along the flight trajectory. Unless the flight plan was designed to allow for multiple transects of a single cloud, including flights at CBH and CTH, no statement towards individual cloud dimensions can be made. Additionally, the commencement of a flight has to comply with local flight rules i.e. terrain clearance. Hence, we derived a vertical profile from all individual cloud transects of low-level clouds as a

proxy. Instead of speaking of a single continuous compact cloud layer, one must rather use the notion of different cloud encounter along a multitude of low-level clouds that were compiled in a single vertical profile.

Nevertheless, the used vertical low-level cloud statistics can be considered as representative for low clouds in the considered period.

To highlight this, a comparison between an example of a single day of measured low-level clouds to the statistical vertical profile support the aptness of the derived vertical profiles and their input in the RT model.

The vertical profile of the cloud measurements of July 5 suggest the occurrence of lowlevel clouds between 750 m and 1750 m between 11:30 to 13:50 UTC, with maximum droplet number concentrations of 1200 cm-3. The flight was intended as a comparison flight from Lome to Savé and a turn over the ground station called KITcube. A comparison with the ceilometer at station, a model CHM15k NIMBUS (https://www.lufft.com/dede/produkte/wolken-schneehoehensensoren-306/ceilometer-chm-15k-nimbus-2093/), supports the flight measurements in terms of occurrence of low-level clouds. The ceilometer retrieval (provided by N. Kalthoff) shows the presence of a nocturnal stratiform cloud cover as a ~200 m thick band around 300 m agl during night and the early morning hours, resulting from an interaction of the African easterly jet and the nocturnal low-level jet (Zouzoua et al.,2021). As the sun rises, the CBH lifts and the thickness of the cloud cover increases with a rise in CTH. Around noon, the stratiform clouds break up and clouds are found in the entire range between 700 m agl and 1700 m agl. These results fit to the inflight measurements aboard the DLR Falcon-20 on this specific day, as well as to the vertical profile in the low-level cloud statistics from all research flights.

Figure 1: Representation of a vertical profile of cloud droplet number concentration measured aboard the DLR Falcon 20 research aircraft of low-level clouds from 5 July 2016 on a comparison flight from Lome, Togo to Savé Benin where a ground station was overflown. The representation of the vertical profile shows cloud encounter between 750 m agl and 1750 m agl, with higher concentrations on four distinct altitudes, which correlate to the main cruising altitudes.

---

## Author Comment (AC2)

**"Pollution slightly enhances atmospheric cooling by low-level clouds in tropical West Africa", submitted to ACP by Valerian Hahn et al., 2022**

We thank Referee #2 for his valuable comments and the effort that has been put reviewing the submitted manuscript on the pollution effects on atmospheric cooling by low-level clouds in tropical West Africa.

**Response to the General Comment:**

We thank the Referee for the suggestion and completely agree to add a conclusion section to the manuscript, which outlines the impact and implications of this study.

This includes the discussion of the representativity of the results concerning season and location.

This analysis comprises a data set from the DACCIWA aircraft campaign segment in June and July 2016. In light of a predicted tripling of anthropogenic emissions in tropical West Africa between 2000 and 2030 alongside an increasing urbanization and demographic growth, the overarching goal of the DACCIWA aircraft campaign, as part of a multidisciplinary project, was to quantify the consequences for the local climate. The underlying motivation was to better understand interactions between emissions, clouds, radiation, precipitation, and regional circulations, with an emphasis on a better understanding of the two-way cloud and aerosol impacts on the radiation and energy budgets from the cloud scale to the scale of the West African monsoon circulation with a certain attention to low-level clouds (Knippertz et al., 2013).

With our study we directly address one essential campaign objective by analysing the radiative impact of polluted versus less polluted low-level clouds, which before the campaign were suspected of being highly susceptible towards local emissions (Knippertz et al., 2015).

Knippertz et al. (2017) describe the meteorology and chemistry corresponding to this monsoon onset season, for which the data set at hand is representative.

Closely correlated to this study is the ubiquitous background aerosol, transported from central and south Africa into the measurement region. This is correlated to agricultural land use in southern and central Africa where each year slash-and-burn methods are used for land cultivation. Outside this period (including a certain transport delay) background biomass burning aerosol from these sources vanish.

Nevertheless, this instance allows us to draw conclusions for similar cases, where additional urban emissions are released into an already polluted background environment. Take cities in South America for instance, here, slash-and-burn methods are used in the Amazon Rainforest. A blending with urban emissions from densely populated conurbations likely has comparable implications. The same holds for regions in south east Asia, either in terms of biomass burning from agriculture or in the agglomeration regions of megacities, where there is no seasonality. These Implications will be added as a in a conclusion section to bolster the significance of our study.

**Comment 1:**
Line 25: is accumulation aerosol on a number concentration basis?
**Answer 1:**
Accumulation aerosol has been used in terms of ambient particle number concentrations.
The new draft has been modified accordingly.

**Comment 2:**
Line 31: Add close parenthesis
**Answer 2:**
A parenthesis has been included at this spot in the revised version.

**Comment 3:**
Line 36: I don't understand what is being said in the sentence: "Thus, polluted low-level clouds add only a relatively small contribution on top of the already exerted cooling by low-level clouds in view of a background atmosphere with elevated aerosol loading". Are the authors making the case that the indirect cooling from polluted clouds is similar to the direct cooling from pollution aerosols in the absence of clouds? Please clarify.
**Answer 3:**
The phrasing of this sentence might be inconclusive. We have changed it in the revised version to:
"Thus, the exerted atmospheric cooling by low-level clouds only increases ever so slightly in light of their formation in an environment with a substantial increase of accumulation mode aerosol on top of an already elevated aerosol background."

**Comment 4:**
Line 234: Should this be $\alpha\Delta\lambda$?
**Answer 4:**
This has been accounted for in the revised version.

**Comment 5:**
Line 266: Were the OPC size distributions fitted (say to a lognormal function) in order to account for the accumulation mode contribution below 250 nm?
**Answer 5:**
The particle number concentration described as accumulation mode aerosol used in this study comes entirely from the OPC Instruments with a cut off at 250 nm. No subsequent fitting to a density function has been performed. Although this lower measurement cut off does not consider the entire accumulation mode, this estimate was regarded as sufficient to be correlated to CO as a pollution tracer.

**Comment 6:**
Lines 270-272: Were aerosol measurements made within the vicinity of the cloud? Is the accumulation mode aerosol just below cloud well correlated with the CO-correlation-based estimate?
**Answer 6:**
The aerosol measurements were entirely outside, but (below 1800 m) constantly in the vicinity of clouds. The flight strategy, in order to accommodate all campaign goals, included only few instances where we probed along a prescribed flight track on various altitudes. Unfortunately,

a precise analysis of accumulation mode aerosol measurements just below individual clouds is not possible.

We would like to draw the Referee's attention to the inflight images in figure 1a-c for a better depiction of the cloud situation of low-level clouds land inwards. A typical phenomenon during the campaign was a break-up of the shallow stratiform cloud deck during the late morning and noon, that has formed during the night. Convection either shallow or with a certain vertical extent formed during the afternoon.

Thus, flying below 1800 m altitude necessarily brought us close to clouds (and somewhat below clouds, when flying below).

**Comment 7:**
Figure 5: what is the lowest droplet diameter shown on the x-axis?
**Answer 7:**
We thank the Referee #2 for this hint. The x-axis was readjusted according to the low droplet size threshold of 3 µm.

Knippertz, Peter: Dynamics-aerosol-chemistry-cloud interactions in West Africa- DACCIWA Fact Sheet, https://www.imk-tro.kit.edu/download/Dacciwa-FactSheet-eng.pdf, 2013.

Knippertz, P., Coe, H., Chiu, J. C., Evans, M. J., Fink, A. H., Kalthoff, N., Liousse, C., Mari, C., Allan, R. P., Brooks, B., Danour, S., Flamant, C., Jegede, O. O., Lohou, F., and Marsham, J. H.: The DACCIWA project: Dynamics-aerosol-chemistry-cloud interactions in West Africa, B. Am. Meteorol. Soc., 96, 1451–1460, https://doi.org/10.1175/BAMS-D-14-00108.1, 2015.

Knippertz, P., Fink, A. H., Deroubaix, A., Morris, E., Tocquer, F., Evans, M. J., Flamant, C., Gaetani, M., Lavaysse, C., Mari, C., Marsham, J. H., Meynadier, R., Affo-Dogo, A., Bahaga, T., Brosse, F., Deetz, K., Guebsi, R., Latifou, I., Maranan, M., Rosenberg, P. D., and Schlueter, A.: A meteorological and chemical overview of the DACCIWA field campaign in West Africa in June-July 2016, Atmospheric Chemistry and Physics, 17, 10 893–10 918, https://doi.org/10.5194/acp-17-10893-2017,https://www.atmos-chem-phys.net/17/10893/2017/, 2017.

a)

[Figure]

b)

[Figure]

c)

[Figure]

*Figure 1 a), b), c): Inflight images from the Falcon research aircraft during DACCIWA campaign from different days.*

---

## Author Comment (AC3)

"Pollution slightly enhances atmospheric cooling by low-level clouds in tropical West Africa", submitted to ACP by Valerian Hahn et al., 2022

**Reply Referee# 1**

Dear Reviewer,

Your feedback is greatly appreciated and was helpful in improving the quality of this research. We value the constructive criticism and thoughtful comments, which have helped to identify areas that require further clarification and refinement.

We carefully considered your suggestions and incorporated them into the revised manuscript, as specified below.

**General Comment 1:**
A. Introduction: I don't understand the primary motivation for this study. There are lots of nice details about prior work but no clear story about what it all adds up to and where important knowledge gaps remain. Why is this new work needed? How does it relate to the previous work?

**Reply General Comment 1:**
We thank Referee#1 for the indication to emphasize the motivation of this study. As Knippertz et al. (2015) outline, tropical West Africa faces a major population growth and urbanization. The effect of an increase in anthropogenic emissions is not only of concern in view of health aspects, but leads to uncertainties for future regional climate. Knippertz et al. (2015) raise the question on the susceptibility of clouds to increases in anthropogenic pollution in West Africa.

Previous studies found discrepancies in local weather models, also associated with a misrepresentation of multilayered cloud structures in this region (van der Linden et al., 2015; Birch et al., 2014).

Also, the contribution of low-level clouds to the energy budget of the atmosphere during the monsoon onset phase in West Africa was investigated for which detailed in-situ cloud measurements were required.

Our study can be very well embedded in the body of existing studies and expands their current state of knowledge within the scope of the DACCIWA campaign, such as Haslett et al. (2019), who analyzed the aerosol abundance in western Africa, van der Linden et al. (2015) and Hill et al. (2018), who investigated the occurrence of cloud structures and patterns from remote sensing retrievals and latter calculated an overall radiation budget. While Taylor et al. (2019) focus on a statistical analysis of in-situ cloud data and distinguish between local and continental sources (inland) as well as transported maritime air masses, our study makes a distinction between polluted and less polluted clouds, based on CO mixing ratios, which we link to accumulation mode aerosol. Also, Taylor et al. (2019) use a parcel model to explain the formation and the observed difference between maritime and inland low-level clouds.

Still, the role of polluted and less polluted low-level clouds for the regional radiation budget is not answered. Here we use radiative transfer calculations with the measured low-level clouds to investigate the impact of pollution on clouds and the radiation budget. We derive instantaneous radiative forcing and heating rates, as a first step to quantify the effect of increased aerosol emissions on low-level clouds in tropical West Africa.

We changed the manuscript as follows:

Abstract lines 34-40 (revised MS):
Radiative transfer simulations show a non-negligible influence of higher droplet number concentrations and smaller particle sizes on the net radiative forcing at the top of atmosphere of -16.3 W/m² of  polluted with respect to  less polluted clouds and lead to a change in instantaneous heating rates of -18 K day$^{-1}$ at the top of  clouds at noon.  Thus, the atmospheric cooling by low-level clouds increases only slightly in the polluted case due to the already elevated background aerosol concentrations.

Lines 105-113 (revised ms): ~~In our study we calculate the radiative impact of inland continental low-level clouds in West Africa based on a comprehensive 110 data set of in-situ observations from the 12 measurement flights of the Falcon 20 research aircraft during the DACCIWA airborne campaign. Additionally, we simulate how the increased anthropogenic pollution of low-level clouds affects the radiation budget and in which direction such effects could change the local climate.~~  Here we investigate the impact of low-level clouds to the energy budget during the monsoon onset phase in West Africa. We calculate the instantaneous radiative impact of inland continental low-level clouds in West Africa based on a comprehensive data set of in-situ observations from the 12 measurement flights of the Falcon 20 research aircraft during the DACCIWA airborne campaign. Based on the measurements we simulate the impact of increased anthropogenic pollution on low-level clouds, the atmospheric radiation budget and heating rates.

Lines 446-448 (revised ms): This data set contributes to fill the gap of scarce cloud measurements in  the Gulf of Guinea ranging  from Benin to the east and Côte d'Ivoire in the west. We also investigate effects of pollution on low-level clouds and on the radiation budget in this region.

Lines 493-500 (revised ms): The growing economy and ongoing urbanisation of the West African suggest further increases in aerosol emissions and related changes in cloud condensation nuclei concentrations in future. Still our study suggests that clouds have a lower susceptibility to aerosol in a regime with high background aerosol concentrations and that related cloud radiative forcing are small and might be damped by a medium-high and high clouds. Results of this study are representative for the monsoon onset period in West Africa, associated with long-range transport of biomass burning aerosol related to agricultural land use in southern and central Africa where each year slash-and-burn methods are used for land cultivation. It remains to be investigated whether the results from this study can be transferred to other regions in the world with higher pollution levels.

**General Comment 2:**
B. Cloud adjustments and radiative forcing: It would be helpful to be more precise in the discussion here. What you are calculating is the instantaneous radiative forcing due to the Twomey effect alone (effect of greater aerosol leading to greater cloud droplet number concentration and smaller effective radius for the same amount of cloud water). It neglects both cloud adjustments to the Twomey effect and other atmospheric adjustments to the changes in heating profiles and thus is not the effective radiative forcing, which is a distinction worth pointing out. It also may be worth thinking about the effect of aerosols on the clear-sky fluxes, as the "clean" and "polluted" cases would presumably also have different AOD values.
**Reply General Comment 2:**

The distinction between an instantaneous radiative forcing and the effective radiative forcing as pointed out by Referee#1 is important. We have reviewed the manuscript in order to make sure to use the terms instantaneous net radiative forcing and instantaneous heating rates throughout the entire manuscript correctly. The investigation of cloud and atmospheric adjustments according to the demonstrated aerosol-cloud interaction cannot be simulated in the discrete RT-model libRadtran and lies outside the scope of our study. Analyses of adjustment mechanisms with data from the DACCIWA period were analyzed e.g. by Pante et al. (2021), who find a likely cause of reduced precipitation caused by enhanced aerosol levels, but a cloud lifetime effect, such as shown by Christensen et al. (2020) has not been analyzed. Nevertheless, cloud and ongoing atmospheric adjustments are already implicitly included in the measured data set. Just like Douglas and L'Ecuyer (2020) a determination and quantification of cloud adjustment processes need a more holistic approach and might feature a synopsis of additional data sources and measurements to in-situ data alone, such as remote sensing and cloud microphysical models. The question of a varying AOD is interesting in the context of determining effective radiative forcing when quantifying of the influence of local sources versus aerosol sources from long-range transport. However, our study specifically examines the influence of a ubiquitous aerosol background and its influence on cloud microphysics and the mere aerosol-cloud effect on instantaneous radiative forcing.

We changed the manuscript as follows:

(revised ms): Throughout the ms we use the term instantaneous heating rates, instantaneous radiative effects and instantaneous net radiative forcing.

Lines 74-77 (revised ms): Interactions of cloud condensation nuclei (CCN) from biomass burning aerosol and low-level clouds have also been studied by previous studies (Painemal et al., (2014); Douglas and L'Ecuyer, 2020; Christensen et al., 2020; Menut et al., 2018; Kaufman and Fraser, 1997; Kaufman et al., 2005; Ramanathan et al., 2001)

Lines 365-377 (revised ms):
An aerosol effect on clouds and its implications for the radiation budget (Twomey, 1991) could lead to an increase in cloud droplet number concentration, at the same time reducing the effective radius, given that the liquid water content is similar (since LWC is basically given by the amount of condensable water and thus by the updraft speed).
As Panicker et al. (2010) describe, the origin and long-range transport of aerosol, as is the case in West Africa during the campaign, could play a role in the formation of a positive or negative relationship between aerosol number and ED. A saturation and eventual reversal of the Twomey effect is observed in Wang et al. (2015) at AOTs of 0.4 to 0.5 and above. A mean AOT of 0.38 from the Aeronet data might explain the comparatively small difference in effective diameters between polluted and less polluted low-level clouds in our study. Qiu et al. (2017) suggest that the precipitable water vapor and increasing collision-coalescence moderates the relationship between AOT and ED, which may lead to an anti-Twomey effect. Radiative effects resulting from increased aerosol loading can also lead to a reversal of the relationship through entrainment and coalescence (Kathri et al., 2022). The Twomey effect identified in our study is used as a basis to derive the instantaneous cloud radiative forcing and the instantaneous heating rates based on both, greater CDNC and smaller ED, while neglecting any other adjustment effects.

Lines 483-487 (revised ms): In order to determine an effective radiative forcing based on instantaneous heating rates, further studies of cloud and atmospheric adjustment are needed using a thermodynamic model that also accounts for adjustment of cloud

microphysics. The adjustments according to the demonstrated aerosol-cloud interaction and corresponding radiative response cannot be simulated by use of a one-dimensional RT-model like the UVSPEC/DISORT routine from libRadtran and the calculation is outthe scope of our study.

**General Comment 3:**
C. Cloud specification in RT model: In general, does the ~1 km cloud make sense? Is that how thick the clouds typically are? It would be helpful to perhaps discuss distributions of cloud top and base heights for the polluted and clean cases. This would also be relevant for thinking about how to interpret the vertical profiles in Figure 6 (see comment below).

**Reply General Comment 3:**
We thank the Referee for the suggestion to include a discussion of the distribution of cloud top and cloud base heights in the manuscript.

The determination of the vertical cloud structure is a representation of a height averaged profile over all campaign flights with the DLR-Falcon performed around noon. They show a cloud base height (CBH) at an average of 666.29 ±82.82 m and cloud top height (CTH) around 1909.40 ±440.12 m. While interpreting the vertical profile, one has to consider the underlying measuring and flight principles inherent in aircraft-based in-situ measurements: Contrary to genuine vertical profiles e.g. from balloon soundings, aircraft measurements incorporate a much larger horizontal component. An aircraft based vertical atmospheric survey only features measurements from altitudes along the flight trajectory. Unless the flight plan was designed to allow for multiple transects of a single cloud, including flights at CBH and CTH, no statement towards individual cloud dimensions can be made. Additionally, the commencement of a flight has to comply with local flight rules i.e. terrain clearance. Hence, we derived a vertical profile from all individual cloud transects of low-level clouds as a proxy. Instead of speaking of a single continuous compact cloud layer, one must rather use the notion of different cloud encounter along a multitude of low-level clouds that were compiled in a single vertical profile.

Nevertheless, the used vertical low-level cloud statistics can be considered as representative for low clouds in the considered period.

The comparison between ceilometer measurements and airborne in-situ measurements above and in the vicinity of the ground station from 5 July validates the extent of the derived vertical cloud profiles. The ceilometer retrieval (provided by N. Kalthoff) shows the presence of a nocturnal stratiform cloud cover as a ~200 m thick band around 300 m agl during night and the early morning hours, resulting from an interaction of the African easterly jet and the nocturnal low-level jet (Zouzoua et al.,2021). As the sun rises, the

[Figure]

CBH lifts and the thickness of the cloud cover increases with a rise in CTH. Around noon, the stratiform clouds break up and clouds are found in the entire range between 700 m agl and 1700 m agl. These results fit to the inflight measurements aboard the DLR Falcon 20 on this specific day, as well as to the vertical profile in the low-level cloud statistics from all research flights.

**CDNC /cm-3**

*Figure 1: Vertical profile of cloud droplet number concentrations measured aboard the DLR Falcon 20 research aircraft on 5 July 2016 on a comparison flight from Lome, Togo, to Savé, Benin, where a ground station was overflown. The representation of the vertical profile shows cloud encounters between 750 m agl and 1750 m agl, with higher concentrations on four distinct altitudes, which correlate to the main cruising altitudes.*

[Figure]

*Figure 2: Ceilometer retrieval from the KITcube station in Savé, Benin from 5 July 2016. The retrieval shows the evolution of low-level clouds, starting from a nightly stratiform cloud deck that lifts after sunrise, broadens and breaks up around noon. Credits: N. Kalthoff-DACCIWA presentation 2017.*

To give an idea about the radiative influence of a standard deviation modified CBH and CTH, sensitivity studies were performed with the DISORT solver in accordance with the conditions published in the study for the profile of a polluted and less polluted cloud.

As a starting point of the of the sensitivity study, the corresponding measurement profiles from the study were used with a solar zenith angle of 17 November at 12:00 local solar time. Assuming a CBH of 0.8 km and a CTH of 1.85 km, the instantaneous delta net radiative forcing is -17.14 $Wm^{-2}$ at the top of atmosphere and -14.58 $Wm^{-2}$ at the ground.

Based on this case, the entire cloud was raised by 450 m, which is approximately the standard deviation of the CTH (as well as lowered by 100 m, which corresponds to the CBH). This results in delta net radiative forcing at TOA of -17.40 $Wm^{-2}$(-17.04 $Wm^{-2}$) and -14.42 $Wm^{-2}$ (-14.61 $Wm^{-2}$) at BOA. Hereby it is evident that raising (lowering) the vertical

profile derived from the measurements by the standard deviation does not significantly change the instantaneous net radiative forcings caused by the Twomey effect.

Only for the case that the total cloud is stretched by 450 m, so that the CBH is held at 0.8 km but the CTH is now at 2.3 km, results in a delta net radiative forcing at TOA of 12.98 Wm$^{-2}$ and -9.51 Wm$^{-2}$ at BOA. Only with a change of the total optical thickness of the cloud the delta in the instantaneous radiative forcing increases. This, however, means that its completely different cloud, which no longer corresponds to the measurements from the DACCIWA flight campaign.

We changed the manuscript as follows:
Lines 206-207 (revised ms): The low-level clouds measure around noon are compiled into a one-layer surrogate, which is embedded between 790 m and 1870 m and discretizised into 60 model layers.

Lines 345-348 (revised ms):
The determination of the vertical cloud structure is a representation of a height-bin averaged profile over all campaign flights with the DLR-Falcon performed around noon. They show a cloud base height (CBH) at an average of 666 ±83 m and cloud top height (CTH) around 1909 ±440 m. Instead of a single continuous compact cloud layer, different cloud encounters of different low-level clouds were compiled into a single vertical profile.

**Specific Comments:**

**Comment 1:**
Line 71 (old ms): What is meant by "large mode"? A large fraction of the accumulation mode?
**Reply 1:**
Yes, exactly. This statement refers to the observation by Haslett et al. (2019, their figure 3) that in all probed airmasses during the DACCIWA aircraft campaign, classified as upwind marine, continental background and urban outflow the aerosol particle number concentration of accumulation aerosol seems to be somewhat comparable (Fig. 1). This finding, including an assessment of the chemical composition of the probed air masses, led them to the conclusion that the entire region is already influenced by advected biomass burning aerosol from remote sources, likely from agricultural fires in southern and central Africa.  For a clearer understanding the wording of this sentence will be modified accordingly in the revised version of the manuscript.

Lines 72-74 (revised ms): They find a large contribution of accumulation mode aerosol from biomass burning aerosol transported from the southern hemisphere on the large mode of in the background aerosol distribution in West Africa, which acts as cloud condensation nuclei.

**Comment 2:**
Lines 72-73: There are many more studies than just Painemal et al. (2014) that study the effect of smoke CCN on low level clouds! Is there a particular point you want to make here about that paper?
**Reply 2:**
There is no specific intent in mentioning just Painemal et al. (2014). Further citations can be drawn from Kaufman and Fraser, 1997; Kaufman et al., 2005; Ramanathan et al., 2001; Douglas and L'Ecuyer, 2020; Christensen et al., 2020; Menut et al., 2018, just to mention a few. The references have been included in the revised version of the manuscript.

Lines 74-77 (revised ms): Interactions of cloud condensation nuclei (CCN) from biomass burning aerosol and low-level clouds have also been studied investigated by previous

studies (Painemal et al., 2014; Douglas and L'Ecuyer, 2020; Christensen et al., 2020; Menut et al., 2018; Kaufman and Fraser, 1997; Kaufman et al., 2005; Ramanathan et al., 2001).

**Comment 3:**
Line 72: CCN has not yet been defined.
**Reply 3:**
We now define cloud condensation nuclei (CCN) in line 75 of the revised ms.

**Comment 4:**
Line 108: Instantaneous observations are unable to directly measure the "influence of aerosol loading on microphysical properties".
**Reply 4:**
This wording is inaccurate or at least imprecise. We have modified the sentence as follows:
Lines 118-120 (revised ms): Microphysical properties of low-level clouds  were measured in-situ  with the Cloud and Aerosol Spectrometer CAS installed at a wing station of the Falcon 20 research aircraft.

**Comment 5:**
Line 152: Wouldn't CO also trace biomass burning plumes?
**Reply 5:**
This is true. However, the dilution of air masses from central and southern Africa, influenced by biomass burning during the long-range transport, provides the CO background mixing ratio. Additional contributions from local sources such as urban emission plumes, as well as local biomass burning sites, add up to an enhancement of the local CO mixing ratio above the background.
In the revised version of the manuscript, local sources of biomass combustion are considered.
Lines 161-164 (revised ms):
Since  accumulation mode aerosol measurements with the SkyOPC are affected by the presence of clouds, CO concentrations have been used to derive location and dilution of local sources such as urban emission plumes as well as local biomass burning sites (Haslett et al., 2019).

**Comment 6:**
Line 201: Why not give the details of when these clouds were observed here?
**Reply 6:**
We used vertical cloud sequences between 4770 m and 4840 m and 9100 m and 9700 m measured on 6 July 2016 during a climb as input for the mid-level cloud in the RT-simulation. These details are now given in the revised version of the manuscript.

Lines 212-214 (revised ms): LWC/IWC and ED profiles for these clouds have been taken from measurements from 6 July 2016  and are then gradually adjusted for sensitivity studies (section 5.2).

**Comment 7:**
Lines 218-219: Where is the AOD assumed to reside vertically?
**Reply 7:**
We use a standard urban background profile in the lower 2 km taken from Mayer and Kylling (2005) and scale it to the AOT of 0.381 measured by AERONET stations in the region (KITcube Savé, Benin, and Koforidua, Ghana) and campaign period.
Also, we performed a sensitivity study to evaluate the dependence of top of atmosphere net fluxes (SW and LW) on variations in AOT. For a clear sky case at noon a variation of

±20 % of the assumed AOT lead to a change of ±4.9 Wm$^{-2}$. A doubling of the AOT still reduces the TOA net flux by 21.4 Wm$^{-2}$.

Since the focus of our study is on low-level cloud microphysics and resulting radiative properties, an in-depth analysis and variation of the aerosol background was not intended. We are convinced that the combination of a measured AERONET AOT coming from the same region and time period in combination with the selected profile from libRadtran yields a close to realistic representation of aerosol distribution for our low-level cloud RT study.

**Comment 8:**
Line 292: Is the CLARIFY value referring to median CO in the boundary layer? CO in the free tropospheric biomass burning plumes observed during CLARIFY and ORACLES were substantially higher than this value. Also, the median value between clean air and heavily polluted plumes might not be a particularly meaningful metric.

**Reply 8:**
Measurements from the CLARIFY campaign that have been performed directly within heavily polluted biomass burning plumes from central and southern Africa are significantly higher compared to DACCIWA or ACRIDICON (Wendisch et al.,2016) data. Still, the classification of polluted versus less polluted airmasses or clouds according to the CO mixing ratio which we correlated to accumulation mode aerosol is a valid method.

Line 300-303 (revised ms):
With median CO mixing ratios of 75 ppmv above the South Atlantic Ocean measured directly in biomass burning plumes during the CLARIFY-2017 campaign (Haywood et al., 2021), the measured CO average mixing ratio during DACCIWA is significantly lower, but can still be used to distinguish between polluted and less polluted air parcels .

**Comment 9:**
Line 303: Are these measurements only around noon? This should be clarified in the methods section.

**Reply 9:**
As reported by Taylor et al, (2019) three aircraft were operated during Dacciwa and often the Falcon went second during the day and as a consequence around Falcon measurements were performed around noon between 10 am and 2 pm with earliest to latest Falcon measurements between 9 am and 3 pm. Eventually all flights covered noon.

Line 316-317 (revised ms)
The mean effective diameter of the  low-level less polluted clouds is 14.8 µm and most of the measurements were performed between 10 am and 2 pm.

**Comment 10:**
Lines 323-326: Why is this not shown?

**Reply 10:**
Vertical velocity measurements from the basic instrumentation system aboard the Falcon research aircraft revealed mean updrafts of 0.20 ms$^{-1}$ within clouds, among the discussed data set of polluted and less polluted low-level clouds in West Africa.

Both histograms of observed updraft speeds in low-level clouds for the polluted, as well as the less polluted cases reveal a quite similar behavior, except a slightly broader distribution for the polluted case.

Taylor et al. (2019) assume in their packet model the 3$^{rd}$ quartile as estimate for morning updraft speeds. Following their example in also considering the 75$^{th}$ percentile as updraft speeds we yield a value of w = 0.453 ms$^{-1}$, which fits to Taylor et al. (2019). Analyzing the 75$^{th}$ percentile of the updraft speeds of each of the low-level cloud cases individually reveal a deviation of approximately 10 %.

These assumptions subsequently fit quite well to the modelled (observed) CDNC increase of 31 % (26 %) between polluted and less polluted low-level clouds, as also calculated by Taylor et al. (2019).
We now include Figure xx in the revised manuscript and added the following text:
Changes in the revised ms:

Lines 333-342 (revised ms): ~~The mean updraft of 22 cm/s indicates low updraft velocities in the noontime within low-level clouds. There is no statistically significant difference in updraft speed distributions for the high and less polluted clouds. For our study, differences in updraft speed distributions are small for the polluted and less polluted case and have a minor influence on changes in cloud particle size distributions.~~
As shown in Figure 5, the difference between the two vertical velocity distributions is minor, as peak relative frequency of occurrence differs by 1.5%. The median vertical velocities of 0.01 m s$^{-1}$ (-0.09 m s$^{-1}$) in polluted (less polluted) low-level clouds indicate no discernible overall trend in vertical motion during measurements, with no statistically significant difference in updraft speed distributions for high and less polluted clouds. In both cases, the standard deviations are 0.69 m s$^{-1}$ (0.62 m s$^{-1}$), with maximum vertical velocities reaching 4 m s$^{-1}$ (3 m s$^{-1}$). Since differences in updraft speed distributions are small for the polluted and less polluted cases in our study, we assume that it has only a minor influence on changes in cloud particle size distributions.

We included Figure 5 in the revised ms.

[Figure]

*Figure 5: Histograms of vertical velocities for the polluted a) and less polluted b) low-level clouds show an overall similar distribution, with median updraft speeds close to 0 ms-1 and standard deviations lower than 0.69 ms-1 in both cases.*

.

**Comment 11:**
Figure 6: The droplet size results are potentially convolving cloud vertical structure and microphysical differences. An adiabatic cloud should have increasing droplet size with height. Are the distributions of cloud tops and bases similar between the more and less polluted cases? If not, that would influence the comparison here.
**Reply 11:**
In Figure 7 (revised ms) we also show the vertical profile of cloud properties and the expected increase in ED. The distributions of cloud tops and bases have been discussed in General Comment 3 and they are similar for more and less polluted clouds as the Falcon measurements were made around noon. There is a diurnal evolution of the clouds as discussed by Taylor et al., (2019).

**Comment 12:**
Lines 344-347: This is a somewhat confusing and oversimplified discussion of indirect aerosol effects. Cloud adjustments to the Twomey effect (holding LWC constant, greater aerosol leads to greater CDNC/lower ED) can be large in magnitude and substantially enhance or counteract the radiative forcing from the Twomey effect alone. Your study only addresses the Twomey effect, but the neglect of adjustments should be mentioned as a source of uncertainty.

**Reply 12:**
We thank the Referee for this valuable hint. An improved discussion on the indirect aerosol effect the is included in the revised version of the ms.

Lines 365-377(revised ms):
An aerosol effect on clouds and its implications for the radiation budget (Twomey, 1991) could lead to an increase in cloud droplet number concentration, at the same time reducing the effective radius, given that the liquid water content is similar (since LWC is basically given by the amount of condensable water and thus by the updraft speed).
As Panicker et al. (2010) describe, the origin and long-range transport of aerosol, as is the case in West Africa during the campaign, could play a role in the formation of a positive or negative relationship between aerosol number and ED. A saturation and eventual reversal of the Twomey effect is observed in Wang et al. (2015) at AOTs of 0.4 to 0.5 and above. A mean AOT of 0.38 from the Aeronet data might explain the comparatively small difference in effective diameters between polluted and less polluted low-level clouds in our study. Qiu et al. (2017) suggest that the precipitable water vapor and increasing collision-coalescence moderates the relationship between AOT and ED, which may lead to an anti-Twomey effect. Radiative effects resulting from increased aerosol loading can also lead to a reversal of the relationship through entrainment and coalescence (Kathri et al., 2022). The Twomey effect identified in our study is used as a basis to derive the instantaneous cloud radiative forcing and the instantaneous heating rates based on both, greater CDNC and smaller ED, while neglecting any other adjustment effects.

**Comment 13:**
Line 361: Twomey effect only, not "pollution effect," which could encompass direct, semidirect, and indirect effects.

**Reply 13:**
Changed accordingly in the modified manuscript.

Line 394-395 (revised ms): The net forcing, which, as defined in this study, is based on the  Twomey effect only, decreases with decreasing cloud cover.

**Comment 14:**
Lines 379-381: I don't follow where this discussion is coming from.

**Reply 14:**
Taking the net radiative forcing for the low-level cloud case, integrated over 24 hours yields RFnet = -3.9 W m$^{-2}$. Under consideration of an additional medium level cloud with a COT=3.1 averaged over 24h yields a net forcing at top of atmosphere of RFnet = -4.0 W m$^{-2}$. Increasing the COT of the medium-level cloud, as done in our sensitivity study, increases the 24h averaged net forcing of the two-cloud-layer case, thus has a greater impact on the net forcing at TOA by our low-level clouds alone. Vice versa a smaller coverage (small COT) of the medium- level cloud over the homogeneous boundary layer cloud has less impact on the net forcing at TOA as exerted by low level-clouds alone.

Formulations have been changed and the last sentence "…; conversely, a homogeneous medium level cloud …" has been deleted because this has not been studied and corresponding numbers from radiative transfer calculations are not available.

Lines 413-417 (revised MS): These numbers are valid for a horizontally homogeneous medium-high cloud with 100 % coverage. Note, a smaller coverage of  medium-high cloud over the homogeneous boundary layer cloud would have  less dampening impact on the net forcing at TOA due to a reduced effective optical thickness.

**Comment 15:**
Lines 398-399: The SW rate isn't converging to the LW values, the total is converging to LW, right? Wouldn't it just be easier to say the SW rate approaches zero?
**Reply 15:**
Thank you, the sentence has been changed.

Line 429-430 (revised ms): With increasing SZA, the SW warming decreases approaching to zero , at SZA ≤ 0 only  LW cooling is effective .

**Comment 16:**
Line 413: Lower than? Not "smaller."
**Reply 16:**
Thanks, this has been changed in the modified manuscript.

Line 448-450 (revised ms): The characterisation of low-level clouds (below 1800 m altitude) over this region, between 29 June and 14 July 2016 shows median cloud droplet number concentrations around 270 $cm^{-3}$, as measured from the CAS underwing probe.

**Comment 17:**
Line 427: "For an entire day" is ambiguous here, as the net forcing is positive at night. Integrated over an entire day?
**Reply 17:**
Thank you for pointing this out. We integrate over the day and not "for an entire day". Changed accordingly in the modified manuscript to "integrated over the day".

Lines 462-464 (revised ms): Seeding the simulation with the microphysical properties from the in-situ measurements shows a negative radiative net forcing at TOA ($F_{net, SW+LW, CO \geq 155\ ppbv}$ minus $F_{net, SW+LW, CO \leq 135\ ppbv}$ ) integrated over  the day for the low-level boundary layer cloud.

**Comment 18:**
Line 446: I'm not sure how you're using "climate sensitivity" in this sentence.
**Reply 18:**
Climate sensitivity is not the correct wording here. We changed the sentence to:

Lines 490-496 (revised ms):
The growing economy and ongoing urbanisation in West Africa suggest further increases in aerosol emissions and related changes in cloud condensation nuclei concentrations in

future. Still, our study suggests that clouds have a lower susceptibility to aerosol in a regime with high background aerosol concentrations and that related instantaneous cloud radiative forcing changes are small and might be damped by a medium-high and high clouds.

**Comment 19:**
Forcing values aren't accounting for any change in AOD
**Reply 19:**
This is correct. Although AOD measurements from the Aeronet measurement network were taken from July 2016, for the radiative transfer calculations there has been no variations of AOD involved in this study, which solely focused on the contribution of low-level clouds.

---

## Author Comment (AC4)

"Pollution slightly enhances atmospheric cooling by low-level clouds in tropical West Africa", submitted to ACP by Valerian Hahn et al., 2022

**Reply Referee#2**

Dear Reviewer,

Your feedback is greatly appreciated and was helpful in improving the quality of this research. We value the constructive criticism and thoughtful comments, which have helped to identify areas that require further clarification and refinement.

We carefully considered your suggestions and incorporated them into the revised manuscript, as specified below.

**General Comment:**

The manuscript presents a report of cloud measurements conducted during DACCIWA, which are classified as either clean or polluted clouds, and then used as inputs to radiative transfer calculations to estimate TOA radiative forcing across the cases. The results are consistent with the first indirect effect -- there is more TOA net radiative cooling for the polluted cloud cases than for the clean cloud cases (as estimated by CO). While this is a nice summary of some of the campaign measurement results with extension to radiative forcing via simple calculations, the scientific depth of analysis is rather shallow. What are we to conclude from this campaign with regard to aerosol-cloud interactions, and would we expect that the results of this interesting set of cases from June-July, 2016, would be regionally representative or consistent with other seasons or years? A good first step would be to add a conclusions section to the manuscript that sums up the impact and implications of the campaign results. Overall, the manuscript is well written and the methods and results sections are appropriately detailed. It is appropriate for ACP. I'd recommend the manuscript be revised to bolster the conclusions and to address the minor comments below:

**Reply to the General Comment:**

We thank the Referee for the suggestion and completely agree to add a conclusion section to the manuscript, which outlines the impact and implications of this study.

This includes the discussion of the representativity of the results concerning season and location.

This analysis comprises a data set from the DACCIWA aircraft campaign segment in June and July 2016. In light of a predicted tripling of anthropogenic emissions in tropical West Africa between 2000 and 2030 alongside an increasing urbanization and demographic growth, the overarching goal of the DACCIWA aircraft campaign, as part of a multidisciplinary project, was to quantify the consequences for the local climate. The underlying motivation was to better understand interactions between emissions, clouds, radiation, precipitation, and regional circulations, with an emphasis on a better understanding of the two-way cloud and aerosol impacts on the radiation and energy budgets from the cloud scale to the scale of the West African monsoon circulation with a certain attention to low-level clouds (Knippertz et al., 2013).

With our study we directly address one essential campaign objective by analysing the radiative impact of polluted versus less polluted low-level clouds, which before the campaign were suspected of being highly susceptible towards local emissions (Knippertz

et al., 2015). Knippertz et al. (2017) describe the meteorology and chemistry corresponding to this monsoon onset season, for which the data set at hand is representative.

Closely correlated to this study is the ubiquitous background aerosol, transported from central and south Africa into the measurement region. This is correlated to agricultural land use in southern and central Africa where each year slash-and-burn methods are used for land cultivation. Outside this period (including a certain transport delay) background biomass burning aerosol from these sources vanish.

Nevertheless, this instance allows us to draw conclusions for similar cases, where additional urban emissions are released into an already polluted background environment. Take cities in South America for instance, here, slash-and-burn methods are used in the Amazon Rainforest. A blending with urban emissions from densely populated conurbations likely has comparable implications. The same holds for regions in south east Asia, either in terms of biomass burning from agriculture or in the agglomeration regions of megacities, where there is no seasonality. These Implications will be added as in a conclusion section to bolster the significance of our study.

**Changes in accordance to the general comment:**

Lines 105-113 (revised ms): ~~In our study we calculate the radiative impact of inland continental low-level clouds in West Africa based on a comprehensive 110 data set of in-situ observations from the 12 measurement flights of the Falcon 20 research aircraft during the DACCIWA airborne campaign. Additionally, we simulate how the increased anthropogenic pollution of low-level clouds affects the radiation budget and in which direction such effects could change the local climate.~~ Here we investigate the impact of low-level clouds to the energy budget during the monsoon onset phase in West Africa. We calculate the instantaneous radiative impact of inland continental low-level clouds in West Africa based on a comprehensive data set of in-situ observations from the 12 measurement flights of the Falcon 20 research aircraft during the DACCIWA airborne campaign. Based on the measurements we simulate the impact of increased anthropogenic pollution on low-level clouds, the atmospheric radiation budget and heating rates.

Lines 446-448 (revised ms): This data set contributes to fill the gap of scarce cloud measurements in  the Gulf of Guinea ranging  from Benin to the east and Côte d'Ivoire in the west. We also investigate effects of pollution on low-level clouds and on the radiation budget in this region

Lines 456-460 (revised ms): Although a high biomass burning aerosol background entrained from central Africa reduces the delta between both classes and reduces the susceptibility to additional local emissions, an aerosol-cloud effect is still visible in cloud microphysical properties, showing effective droplet diameter between 12.3 µm in the polluted case, compared to an ED of 14.8 µm in the less polluted case and CDNCs that almost double (median) in the polluted case.

Lines 497-500 (revised ms): Results of this study are representative for the monsoon onset period in West Africa, associated with long-range transport of biomass burning aerosol related to agricultural land use in southern and central Africa where each year slash-and-burn methods are used for land cultivation. It remains to be investigated whether the results from this study can be transferred to other regions in the world with higher pollution levels.

**Comment 1:**
Line 25: is accumulation aerosol on a number concentration basis?
**Reply 1:**
Accumulation aerosol has been used in terms of ambient particle number concentrations. The new draft has been modified accordingly.

Line 22-25 (revised ms): Clouds below 1800 meter altitude, identified as boundary layer clouds, were classified according to their carbon monoxide (CO) pollution level into pristine and less polluted clouds (CO < 135 ppbv) and polluted low-level clouds (CO > 155 ppbv) as confirmed by the linear CO to accumulation aerosol number concentration correlation.

**Comment 2:**
Line 31: Add close parenthesis
**Reply 2:**
A parenthesis has been included at this spot in the revised version in line 31 (new ms).

**Comment 3:**
Line 36: I don't understand what is being said in the sentence: "Thus, polluted low-level clouds add only a relatively small contribution on top of the already exerted cooling by low-level clouds in view of a background atmosphere with elevated aerosol loading". Are the authors making the case that the indirect cooling from polluted clouds is similar to the direct cooling from pollution aerosols in the absence of clouds? Please clarify.
**Reply 3:**
The phrasing of this sentence might be inconclusive. We have changed it in the revised version to:
Lines 39-41 (revised ms):  Thus, the exerted atmospheric cooling by low-level clouds only increases ever so slightly in light of their formation in an environment with a substantial increase of accumulation mode aerosol on top of an already elevated background aerosol concentration.

**Comment 4:**
Line 234: Should this be α∆λ?
**Reply 4:**
This has been accounted for in the revised version (line 246 new ms).

**Comment 5:**
Line 266: Were the OPC size distributions fitted (say to a lognormal function) in order to account for the accumulation mode contribution below 250 nm?
**Reply 5:**
The particle number concentration described as accumulation mode aerosol used in this study comes entirely from the OPC Instruments with a cut off at 250 nm. No subsequent fitting to a density function has been performed. Although this lower measurement cut off does not consider the entire accumulation mode, this estimate was regarded as sufficient to be correlated to CO as a pollution tracer.

**Comment 6:**
Lines 270-272: Were aerosol measurements made within the vicinity of the cloud? Is the accumulation mode aerosol just below cloud well correlated with the CO-correlation-based estimate?
**Reply 6:**
The aerosol measurements for the CO-correlation were entirely outside, but constantly in the vicinity of clouds. The flight strategy, in order to accommodate all campaign goals, included only few instances where we probed along a prescribed flight track on various altitudes. Unfortunately, a precise analysis of accumulation mode aerosol measurements just below individual clouds is not possible.

Due to the break-up of the shallow stratiform cloud deck during the late morning and noon, flying below 1800 m altitude necessarily involved sampling in the immediate vicinity of low-level clouds.

**Comment 7:**
Figure 5: what is the lowest droplet diameter shown on the x-axis?
**Reply 7:**
We thank the Referee #2 for this hint. The x-axis was readjusted according to the low droplet size threshold of 3 µm (Figure 5 new MS) and a remark was made in the figure description in Line 325 (new MS).

---

## Author Response (AR1)

**"Pollution slightly enhances atmospheric cooling by lowlevel clouds in tropical West Africa", submitted to ACP by**
**Valerian Hahn et al., 2022**

**Reply Referee#1**

We thank Referee #1 for his valuable comments and the effort that has been put in reviewing the submitted manuscript on the pollution effects and atmospheric cooling by low-level clouds in tropical West Africa.

**General Comment 1:**
A. Introduction: I don't understand the primary motivation for this study. There are lots of nice details about prior work but no clear story about what it all adds up to and where important knowledge gaps remain. Why is this new work needed? How does it relate to the previous work?

**Answer General Comment 1:**
We thank Referee#1 for the indication to emphasize the motivation of this study. To get a better understand of the basic idea for the analysis of the microphysics of low-level clouds in West Africa, one has to see the underlying scientific questions tied to the design of the DACCIWA campaign. As Knippertz et al. (2015) outline tropical West Africa faces a major population growth and urbanization. The effect of an increase in anthropogenic emissions is not only of concern in view of health aspects, but poses an uncertainty for future local climate. Knippertz et al. (2015) suggest that clouds above West Africa could be 'highly susceptible to increases in anthropogenic pollution'.

Previous studies found discrepancies in local weather models, also associated with a misrepresentation of multilayered cloud structures in this region.

The contribution of low-level clouds to the energy budget during the monsoon onset phase in West Africa was a significant scientific question to be answered by the DACCIWA campaign. This is where a knowledge gap was identified by the DACCIWA consortium und our study aims at contributing to fill this gap.

Our study can be very well embedded in the body of existing studies and expands their current state of knowledge within the scope of the DACCIWA campaign, such as Haslett et al. (2019), who analyzed the aerosol provenance, van der Linden et al. (2015) and Hill et al. (2018), who investigated the occurrence of cloud structures and patterns from remote sensing retrievals and latter calculated an overall radiation budget. Whereas Taylor et al. (2019) focus on a statistical analysis of in-situ cloud data and distinguish between local and continental sources (inland) as well as transported maritime air masses, our study makes a distinction between polluted and less polluted clouds, based on CO mixing ratios, which we can reference to accumulation mode aerosol. Furthermore, Taylor et al. (2019) explained with a sensitivity study using a parcel model the observed difference between maritime and inland low-level clouds.

What is left unanswered to this point is the role of polluted and less polluted low-level clouds within the regional radiation budget. Our study extends the current body of literature by adding radiative transfer calculations fed with a derived representation of measured low-level clouds under consideration of the degree of pollution involved in air associated with the genesis of and/or entrainment in low-level clouds. We look at how the Twomey effect influences the instantaneous radiative forcing and instantaneous heating rates. Although we cannot simulate and evaluate overall cloud adjustments with our RT model, in order to come up with an effective radiative forcing, this still is a crucial step

towards classifying the microphysical and radiative influence of increased aerosol emissions on low-level clouds during the monsoon onset period in tropical West Africa.

A greater emphasize on the significance of our study and its aim to answer a major scientific question of the DACCIWA campaign will be included in the manuscript.

**General Comment 2:**
B. Cloud adjustments and radiative forcing: It would be helpful to be more precise in the discussion here. What you are calculating is the instantaneous radiative forcing due to the Twomey effect alone (effect of greater aerosol leading to greater cloud droplet number concentration and smaller effective radius for the same amount of cloud water). It neglects both cloud adjustments to the Twomey effect and other atmospheric adjustments to the changes in heating profiles and thus is not the effective radiative forcing, which is a distinction worth pointing out. It also may be worth thinking about the effect of aerosols on the clear-sky fluxes, as the "clean" and "polluted" cases would presumably also have different AOD values.

**Answer General Comment 2:**
The distinction between an instantaneous radiative forcing and the effective radiative forcing as Referee#1 emphasizes is an important one to be made. Surely, this point should be underlined when discussing the RT model results. We will review the manuscript in order to make sure to use the terms instantaneous net radiative forcing and instantaneous heating rates throughout the entire manuscript correctly. The investigation of cloud and atmospheric adjustments according to the demonstrated aerosol-cloud interaction cannot be simulated in the discrete RT-model libRadtran and lies outside the scope of our study. Analyses of adjustment mechanisms with data from the DACCIWA period were analyzed e.g. by Pante et al. (2021), who find a likely cause of reduced precipitation caused by enhanced aerosol levels, but a cloud lifetime effect, such as shown by Christensen et al. (2020) has not been analyzed. Nevertheless, cloud and ongoing atmospheric adjustments are already implicitly included in the measured data set. Just like Douglas and L'Ecuyer (2020) a determination and quantification of cloud adjustment processes need a more holistic approach and might feature a synopsis of additional data sources and measurements to in-situ data alone, such as remote sensing and microphysical models. The question of a varying AOD is interesting in the context of determining effective radiative forcing when quantifying of the influence of local sources versus aerosol sources from long-range transport. However, our study specifically examines the influence of a ubiquitous aerosol background and its influence on cloud microphysics and the mere aerosol-cloud effect on instantaneous radiative forcing.

This study investigates a significant open research question posed by the DACCIWA consortium prior to the campaign, and extends other studies by performing radiative transfer calculations.

Our findings provide an excellent starting point for further model studies, e.g. to determine an effective radiative forcing of the overall atmospheric system in the region.

**General Comment 3:**
C. Cloud specification in RT model: In general, does the ~1 km cloud make sense? Is that how thick the clouds typically are? It would be helpful to perhaps discuss distributions of cloud top and base heights for the polluted and clean cases. This would also be relevant for thinking about how to interpret the vertical profiles in Figure 6 (see comment below).

**Answer General Comment 3:**
We thank the Referee for the suggestion to include a discussion of the distribution of cloud top and cloud base heights in the manuscript.

The determination of the vertical cloud structure is a representation of a height averaged profile over all campaign flights with the DLR-Falcon, including cloud base height (CBH) at an average of 666.29 ±82.82 m and cloud top height (CTH) around 1909.40 ±440.12 m. While interpreting the vertical profile, one has to consider the underlying measuring and flight principles inherent in aircraft-based in-situ measurements: Contrary to genuine

vertical profiles e.g. from balloon soundings, aircraft measurements incorporate a much larger horizontal component. An aircraft based vertical atmospheric survey only features measurements from altitudes along the flight trajectory. Unless the flight plan was designed to allow for multiple transects of a single cloud, including flights at CBH and CTH, no statement towards individual cloud dimensions can be made. Additionally, the commencement of a flight has to comply with local flight rules i.e. terrain clearance. Hence, we derived a vertical profile from all individual cloud transects of low-level clouds as a proxy. Instead of speaking of a single continuous compact cloud layer, one must rather use the notion of different cloud encounter along a multitude of low-level clouds that were compiled in a single vertical profile.

Nevertheless, the used vertical low-level cloud statistics can be considered as representative for low clouds in the considered period.

To highlight this, a comparison between an example of a single day of measured low-level clouds to the statistical vertical profile support the aptness of the derived vertical profiles and their input in the RT model.

The vertical profile of the cloud measurements of July 5 suggest the occurrence of lowlevel clouds between 750 m and 1750 m between 11:30 to 13:50 UTC, with maximum droplet number concentrations of 1200 cm$^{-3}$. The flight was intended as a comparison flight from Lome to Savé and a turn over the ground station called KITcube. A comparison with the ceilometer at station, a model CHM15k NIMBUS (https://www.lufft.com/dede/produkte/wolken-schneehoehensensoren-306/ceilometer-chm-15k-nimbus-2093/), supports the flight measurements in terms of occurrence of low-level clouds. The ceilometer retrieval (provided by N. Kalthoff) shows the presence of a nocturnal stratiform cloud cover as a ~200 m thick band around 300 m agl during night and the early morning hours, resulting from an interaction of the African easterly jet and the nocturnal low-level jet (Zouzoua et al.,2021). As the sun rises, the CBH lifts and the thickness of the cloud cover increases with a rise in CTH. Around noon, the stratiform clouds break up and clouds are found in the entire range between 700 m agl and 1700 m agl. These results fit to the inflight measurements aboard the DLR Falcon-20 on this specific day, as well as to the vertical profile in the low-level cloud statistics from all research flights.

[Figure]

**CDNC /cm-3**

*Figure 1: Representation of a vertical profile of cloud droplet number concentration measured aboard the DLR Falcon 20 research aircraft of low-level clouds from 5 July 2016 on a comparison flight from Lome, Togo to Savé Benin where a ground station was overflown. The representation of the vertical profile shows cloud encounter between 750 m agl and 1750 m agl, with higher concentrations on four distinct altitudes, which correlate to the main cruising altitudes.*

[Figure]

*Figure 2: Ceilometer retrieval from the KITcube station in Savé, Benin from 5 July 2016. The retrieval shows the evolution of low-level clouds, starting from a nightly stratiform cloud deck that lifts after sunrise, broadens and breaks up around noon. Similarly to the insitu cloud measurements from the aircraft clouds are found on similar altitudes. Credits: N. Kalthoff*

To give an idea about the radiative influence of a standard deviation modified CBH and CTH, sensitivity studies were performed with the DISORT solver in accordance with the conditions published in the study for the profile of a polluted and less polluted cloud.

As a starting point of the of the sensitivity study, the corresponding measurement profiles from the study were used with a solar zenith angle of 17 November at 12:00 local solar time. Assuming a CBH of 0.8 km and a CTH of 1.85 km, the instantaneous delta net radiative forcing is -17.14 $Wm^{-2}$ at the top of atmosphere and -14.58 $Wm^{-2}$ at the ground.

Based on this case, the entire cloud was raised by 450 m, which is approximately the standard deviation of the CTH (as well as lowered by 100 m, which corresponds to the CBH). This results in delta net radiative forcing at TOA of -17.40 $Wm^{-2}$(-17.04 $Wm^{-2}$) and -14.42 $Wm^{-2}$ (-14.61 $Wm^{-2}$) at BOA. Hereby it is evident that raising (lowering) the vertical profile derived from the measurements by the standard deviation does not significantly change the instantaneous net radiative forcings caused by the Twomey effect.

Only for the case that the total cloud is stretched by 450 m, so that the CBH is held at 0.8 km but the CTH is now at 2.3 km, results in a delta net radiative forcing at TOA of 12.98 $Wm^{-2}$ and -9.51 $Wm^{-2}$ at BOA. Only with a change of the total optical thickness of the cloud

the delta in the instantaneous radiative forcing increases. This, however, means that its completely different cloud, which no longer corresponds to the measurements from the DACCIWA flight campaign.

**Specific Comments:**

**Comment 1:**
Line 71: What is meant by "large mode"? A large fraction of the accumulation mode?
**Answer 1:**
Yes, exactly. This statement refers to the observation by Haslett et al. (2019) that in all probed airmasses during the DACCIWA aircraft campaign, classified as upwind marine, continental background and urban outflow the aerosol particle number concentration of accumulation aerosol seems to be somewhat comparable (Fig. 1). This finding, including an assessment of the chemical composition of the probed air masses, led them to the conclusion that the entire region is already influenced by advected biomass burning aerosol from remote sources, likely from agricultural fires in southern and central Africa. For a clearer understanding the wording of this sentence will be modified accordingly in the revised version of the manuscript.

*They find a large contribution of accumulation mode aerosol from biomass burning  transported from the southern hemisphere  in the background aerosol distribution in West Africa, which acts as cloud condensation nuclei.*

[Figure]

**Figure 3.** Size distributions of aerosol in the urban outflow, continental background and upstream marine regimes, measured by the SMPS on board the ATR aircraft. For each regime, the median size distribution is shown by the dark line, the dark shading contains 50 % of the data, and the light shading contains 80 % of the data. The comparison of all three plots in panel **(a)** shows a stable accumulation mode that exists in all three regimes, centred at around 200 nm, while the smaller Aitken mode is much more variable. In panels **(b)**–**(d)**, $N$ shows the median total number concentration summed across the whole distribution, with the lower and upper quartiles shown in brackets; $M$ shows the calculated aerosol mass, assuming an aerosol density of $1.6\,\mathrm{g\,cm^{-3}}$ (Haslett et al., 2019), with the interquartile range again shown in brackets. The Aitken and accumulation modes are labelled in panel **(c)**.

*Figure 3: Aerosol modes from Haslett et al. (2019) divided in upwind marine, continental background and urban outflow, all having in common a similar accumulation mode, which supports the assumption of ubiquitous aged biomass burning aerosol from central and southern Africa.*

**Comment 2:**
Lines 72-73: There are many more studies than just Painemal et al. (2014) that study the effect of smoke CCN on low level clouds! Is there a particular point you want to make here about that paper?

**Answer 2:**
There Is no specific intent in mentioning just Painemal et al. (2014). Further citations can be drawn from Kaufman and Fraser, 1997; Kaufman et al., 2005; Ramanathan et al., 2001; Douglas and L'Ecuyer, 2020; Christensen et al., 2020; Menut et al., 2018, just to mention a few. These will be featured in the revised version of the manuscript.

**Comment 3:**
Line 72: CCN has not yet been defined.

**Answer 3:**
The term cloud condensation nuclei abbreviated as CCN has been introduced in the updated draft.

**Comment 4:**
Line 108: Instantaneous observations are unable to directly measure the "influence of aerosol loading on microphysical properties".

**Answer 4:**
This wording is inaccurate or at least imprecise. We will modify the sentence as follows:
*Microphysical properties of low-level clouds  are measured in-situ  with the Cloud and Aerosol Spectrometer CAS installed at a wing station of the Falcon 20 research aircraft.*

**Comment 5:**
Line 152: Wouldn't CO also trace biomass burning plumes?

**Answer 5:**
This is true. However, the dilution of air masses from central and southern Africa, influenced by biomass burning during the long-range transport, provides the CO background mixing ratio. Additional contributions from local sources such as urban emission plumes, as well as local biomass burning sites, add up to an enhancement of the local CO mixing ratio above the background.
In the revised version of the manuscript, local sources of biomass combustion are considered.
*Since data coverage of accumulation mode aerosol measurements with the SkyOPC is insufficient within clouds, CO concentrations have been used to derive location and dilution of local sources such as urban emission plumes as well as local biomass burning sites, factored by a vast amount of combustion products of organic matter within the plumes (Haslett et al., 2019).*

**Comment 6:**
Line 201: Why not give the details of when these clouds were observed here?

**Answer 6:**
The used proxy for a mid-level cloud was observed on 6 July 2016, before 13 UTC. These details will be given in the modified version.

**Comment 7:**
Lines 218-219: Where is the AOD assumed to reside vertically?

**Answer 7:**
The assumed aerosol profile fed into the RT simulations is a standard profile using the background option with urban in the lower 2 km (Mayer and Kylling, 2005). While taking an AOT of 0.381 from measurements of the Aeronet data from the Region (e.g. Savé, Benin) and campaign period, we assume a representative aerosol profile.
Nevertheless, we performed a sensitivity evaluation to see the dependence of top of atmosphere net fluxes (SW and LW) on variations in AOT. For a clear sky case at noon a

variation of ±20 % of the assumed AOT lead to a change of ±4.9 Wm-2. A doubling of the AOT still reduces the TOA net flux by 21.4 Wm-2.

Since the focus of our study was drawn on low-level cloud microphysics and resulting radiative properties, an in-depth analysis and variation of the aerosol background was not our upmost concern. Still, we are convinced that the combination of a measured Aeronet AOT coming from the same region and time period in combination with the selected profile

[Figure]

*Figure 4: Histograms of vertical velocities for the polluted cases a) and less polluted cases b) show an overall similar distribution.*

from libRadtran yields a close to realistic representation of aerosol distribution for our lowlevel cloud RT study.

**Comment 8:**

Line 292: Is the CLARIFY value referring to median CO in the boundary layer? CO in the free tropospheric biomass burning plumes observed during CLARIFY and ORACLES were substantially higher than this value. Also, the median value between clean air and heavily polluted plumes might not be a particularly meaningful metric.

**Answer 8:**

This objection is correct: Compared to measurements from the CLARIFY campaign that have been performed directly within heavily polluted biomass burning plumes coming from central and southern Africa are in three orders of magnitude higher than what has been measured during the DACCIWA or ACRIDICON campaign. This will be clarified in the updated version of the manuscript. Still, we deem the classification of a polluted versus a less polluted airmass or cloud according to its CO mixing ratio which we can correlate to accumulation mode aerosol as a valid method.

*In light of median CO mixing ratios of 75 ppmv above the South Atlantic Ocean measured directly in biomass burning plumes during the CLARIFY-2017 campaign (Haywood et al., 2021), the measured CO average mixing ratio during DACCIWA is about three orders of magnitude smaller, but can still be used to qualify between polluted and less polluted air parcels .*

**Comment 9:**

Line 303: Are these measurements only around noon? This should be clarified in the methods section.

**Answer 9:**

Although it was no strict strategic flight planning measurement flights were conducted somewhat between 10 o'clock am and 2 o'clock pm. With earliest to latest measurements ranging between 9 o'clock am and 3 o'clock pm. Eventually all flights covered noon. No measurements have been taken during late afternoon, evening, night and early morning hours, thus the term noon-time low-level clouds. This will be elaborated in the modified version.

The mean effective diameter of the  low-level less polluted clouds is 14.8 μm for 90 % of measurements ranging between 10 o'clock am until 2 o'clock pm.

**Comment 10:**
Lines 323-326: Why is this not shown? **Answer 10:**
Vertical velocity measurements from the basic instrumentation system aboard the Falcon research aircraft revealed mean updrafts of 0.20 ms$^{-1}$ within clouds, among the discussed data set of polluted and less polluted low-level clouds in West Africa.
Both histograms of observed updraft speeds in low-level clouds for the polluted, as well as the less polluted cases reveal a quite similar behavior, except a slightly broader distribution for the former.
Taylor et al. (2019) assume in their packet model the 3$^{rd}$ quartile as estimate for morning updraft speeds. Following their example in also considering the 75$^{th}$ percentile as updraft speeds we yield a value of $w = 0.453$ ms$^{-1}$, which fits to Taylor et al. (2019). Analyzing the 75$^{th}$ percentile of the updraft speeds of each of the low-level cloud cases individually reveal a deviation of approximately 10 %.
These assumptions subsequently fit quite well to the modelled (observed) CDNC increase of 31 % (26 %) between polluted and less polluted low-level clouds, as also calculated by Taylor et al. (2019).

**Comment 11:**
Figure 6: The droplet size results are potentially convolving cloud vertical structure and microphysical differences. An adiabatic cloud should have increasing droplet size with height. Are the distributions of cloud tops and bases similar between the more and less polluted cases? If not, that would influence the comparison here.
**Answer 11:**
Cloud bases and cloud tops of both cloud types are comparable, as the clouds emerging from the stratus cover were formed under the same conditions. Observations show that typically for this time of year, deep stratus clouds form overnight in the region, as a result of the transport of cool and humid maritime air inland. The Surface warming throughout the morning causes the nocturnal stratiform clouds to rise and thicken, until their break up during noon. Only towards the afternoon the influence of a convective component becomes more influential (van der Linden et al., 2015; Kalthoff et al., 2018). Figure 3 from Taylor et al. (2019) shows the diurnal cycle of low -level cloud fraction from Satellite observations from SEVIRI during the DACCIWA aircraft campaign, with high cloud fractions between 8 am and 2 pm local solar time over land. Figure 4 shows an inflight image from the DLR Falcon research aircraft of gaps in the former stratiform cloud cover during afternoon break up.

[Figure]

*Figure 5: Cloud fraction of low-level clouds according to solar time and distance from coast in Wet Africa. High cloud fractions between 6 and 14 local solar time are found from the coast line up to 400 km inland. From van der Linden et al. (2015).*

[Figure]

*Figure 6: Representative Inflight image from the Falcon research aircraft during DACCIWA campaign in tropical West Africa from 12 July 2016 around 10:30 am a) above clouds b) below clouds.*

**Comment 12:**

Lines 344-347: This is a somewhat confusing and oversimplified discussion of indirect aerosol effects. Cloud adjustments to the Twomey effect (holding LWC constant, greater aerosol leads to greater CDNC/lower ED) can be large in magnitude and substantially enhance or counteract the radiative forcing from the Twomey effect alone. Your study only addresses the Twomey effect, but the neglect of adjustments should be mentioned as a source of uncertainty.

**Answer 12:**
Adjective "instantaneous" has been added in line 223 as well as in line 349.

**Comment 13:**
Line 361: Twomey effect only, not "pollution effect," which could encompass direct, semidirect, and indirect effects.

**Answer 13:**
Changed accordingly in the modified manuscript.

**Comment 14:**
Lines 379-381: I don't follow where this discussion is coming from.

**Answer 14:**
Taking the net radiative forcing for the low-level cloud case, integrated over 24 hours yields $RF_{net} = -3.9$ W m$^{-2}$. Under consideration of an additional medium level cloud with a COT=3.1 averaged over 24h yields a net forcing at top of atmosphere of $RF_{net} = -4.0$ W m$^{-2}$. Increasing the COT of the medium-level cloud, as done in our sensitivity study, increases the 24h averaged net forcing of the two-cloud-layer case, thus has a greater impact on the net forcing at TOA by our low-level clouds alone. Vice versa a smaller coverage (small COT) of the medium- level cloud over the homogeneous boundary layer cloud has less impact on the net forcing at TOA as exerted by low level-clouds alone.

Formulations have been changed and the last sentence "…; conversely, a homogeneous medium level cloud …" has been deleted because this has not been studied and corresponding numbers from radiative transfer calculations are not available.

**Comment 15:**
Lines 398-399: The SW rate isn't converging to the LW values, the total is converging to LW, right? Wouldn't it just be easier to say the SW rate approaches zero?
**Answer 15:**
Sentence has accordingly been changed

**Comment 16:**
Line 413: Lower than? Not "smaller." **Answer 16:**
Changed accordingly in the modified manuscript.

**Comment 17:**
Line 427: "For an entire day" is ambiguous here, as the net forcing is positive at night. Integrated over an entire day?
**Answer 17:**
Thank you for pointing this out. The objection is correct. Meant here is the integration over the entire day and not "for an entire day". Changed accordingly in the modified manuscript to "integrated over an entire day".

**Comment 18:**
Line 446: I'm not sure how you're using "climate sensitivity" in this sentence.
**Answer 18:**
Climate sensitivity is used to refer to the atmospheric cooling brought on by low-level clouds. As our study demonstrates that this effect does not increase linearly with population growth and ongoing urbanization in tropical West Africa, but is attenuated in the presence of ubiquitous background aerosol.

We agree that discussing about "climate sensitivity" might be far-fetched, when we only regard the Twomey effect alone during the monsoon onset season and discussing net radiative forcings and derived instantaneous heating rates (without regarding a cloud adjustment), without having drawn our conclusions from the effective radiative forcing. The new manuscript will find a more accurate classification. The last two sentences of the "Discussion" have been reworded to make the statements clearer and correct.

**Comment 19:**
Forcing values aren't accounting for any change in AOD
**Answer 19:**
This is correct. Although AOD measurements from the Aeronet measurement network were taken from July 2016, for the radiative transfer calculations there has been no variations of AOD involved in this study, which solely focused on the contribution of low-level clouds.

**Reply Referee#2**

We thank Referee #2 for his valuable comments and the effort that has been put reviewing the submitted manuscript on the pollution effects on atmospheric cooling by low-level clouds in tropical West Africa.

**Response to the General Comment:**
We thank the Referee for the suggestion and completely agree to add a conclusion section to the manuscript, which outlines the impact and implications of this study.
This includes the discussion of the representativity of the results concerning season and location.
This analysis comprises a data set from the DACCIWA aircraft campaign segment in June and July 2016. In light of a predicted tripling of anthropogenic emissions in tropical West Africa between 2000 and 2030 alongside an increasing urbanization and demographic growth, the overarching goal of the DACCIWA aircraft campaign, as part of a multidisciplinary project, was to quantify the consequences for the local climate. The underlying motivation was to better understand interactions between emissions, clouds, radiation, precipitation, and regional circulations, with an emphasis on a better understanding of the two-way cloud and aerosol impacts on the radiation and energy budgets from the cloud scale to the scale of the West African monsoon circulation with a certain attention to low-level clouds (Knippertz et al., 2013).
With our study we directly address one essential campaign objective by analysing the radiative impact of polluted versus less polluted low-level clouds, which before the campaign were suspected of being highly susceptible towards local emissions (Knippertz et al., 2015). Knippertz et al. (2017) describe the meteorology and chemistry corresponding to this monsoon onset season, for which the data set at hand is representative.
Closely correlated to this study is the ubiquitous background aerosol, transported from central and south Africa into the measurement region. This is correlated to agricultural land use in southern and central Africa where each year slash-and-burn methods are used for land cultivation. Outside this period (including a certain transport delay) background biomass burning aerosol from these sources vanish.
Nevertheless, this instance allows us to draw conclusions for similar cases, where additional urban emissions are released into an already polluted background environment. Take cities in South America for instance, here, slash-and-burn methods are used in the Amazon Rainforest. A blending with urban emissions from densely populated conurbations likely has comparable implications. The same holds for regions in south east Asia, either in terms of biomass burning from agriculture or in the agglomeration regions of megacities, where there is no seasonality. These Implications will be added as a in a conclusion section to bolster the significance of our study.

**Comment 1:**

Line 25: is accumulation aerosol on a number concentration basis?

**Answer 1:**

Accumulation aerosol has been used in terms of ambient particle number concentrations. The new draft has been modified accordingly.

**Comment 2:**

Line 31: Add close parenthesis

**Answer 2:**

A parenthesis has been included at this spot in the revised version.

**Comment 3:**

Line 36: I don't understand what is being said in the sentence: "Thus, polluted low-level clouds add only a relatively small contribution on top of the already exerted cooling by low-level clouds in view of a background atmosphere with elevated aerosol loading". Are the authors making the case that the indirect cooling from polluted clouds is similar to the direct cooling from pollution aerosols in the absence of clouds? Please clarify.

**Answer 3:**

The phrasing of this sentence might be inconclusive. We have changed it in the revised version to:

"Thus, the exerted atmospheric cooling by low-level clouds only increases ever so slightly in light of their formation in an environment with a substantial increase of accumulation mode aerosol on top of an already elevated aerosol background."

**Comment 4:**

Line 234: Should this be $\alpha\Delta\lambda$?

**Answer 4:**

This has been accounted for in the revised version.

**Comment 5:**

Line 266: Were the OPC size distributions fitted (say to a lognormal function) in order to account for the accumulation mode contribution below 250 nm?

**Answer 5:**

The particle number concentration described as accumulation mode aerosol used in this study comes entirely from the OPC Instruments with a cut off at 250 nm. No subsequent fitting to a density function has been performed. Although this lower measurement cut off does not consider the entire accumulation mode, this estimate was regarded as sufficient to be correlated to CO as a pollution tracer.

**Comment 6:**

Lines 270-272: Were aerosol measurements made within the vicinity of the cloud? Is the accumulation mode aerosol just below cloud well correlated with the CO-correlation-based estimate?

**Answer 6:**

The aerosol measurements were entirely outside, but (below 1800 m) constantly in the vicinity of clouds. The flight strategy, in order to accommodate all campaign goals, included only few instances where we probed along a prescribed flight track on various altitudes. Unfortunately, a precise analysis of accumulation mode aerosol measurements just below individual clouds is not possible.

We would like to draw the Referee's attention to the inflight images in figure 1a-c for a better depiction of the cloud situation of low-level clouds land inwards. A typical phenomenon during the campaign was a break-up of the shallow stratiform cloud deck during the late morning and

noon, that has formed during the night. Convection either shallow or with a certain vertical extent formed during the afternoon.

Thus, flying below 1800 m altitude necessarily brought us close to clouds (and somewhat below clouds, when flying below).

**Comment 7:**

Figure 5: what is the lowest droplet diameter shown on the x-axis?

**Answer 7:**

We thank the Referee #2 for this hint. The x-axis was readjusted according to the low droplet size threshold of 3 µm.

[Figure]

[Figure]

[Figure]

*Figure 1 a), b), c): Inflight images from the Falcon research aircraft during DACCIWA campaign from different days.*

**Further changes:**

Line 29: Increase in droplet number concentration was 26 % right away instead of the previous 35 %. This value was changes accordingly in the new version.

Line 36: rephrased sentence for a better understanding.

Line 59: added 'and corresponding radiative properties'

Line 73: Include further studies.

Line 100: Defined the scientific gap or motivation as Referee2 suggested.

Line 103: Rephrased for better understanding.

Lines 152-153: rephrased.

Line 187: one-dimensional instead of 1D

Lines 195 & 200: Descriptive addition for a better understanding of the vertical profile as Referee1 suggested.

Line 219: 'Koforidua Anuc (Ghana) and KITcube –Save' instead of 'Koforidua Anuc (Ghana) and KITcube_Save'

Line 344: Increase in droplet number concentration was 26 % right away instead of the previous 35 %. This value was changes accordingly in the new version.

Line 409: Summery and Conclusion instead of Discussion

Line 445ff: included and rephrased this section according to Referee2 suggestions.

---

## Author Response (AR3)

"Pollution slightly enhances atmospheric cooling by low-level clouds in tropical West Africa", submitted to ACP by Valerian Hahn et al., 2022

Dear Referees,

thank you for your thoughtful comments on the draft, which once again helped to further improve the scientific quality and the clarity of our manuscript.

**General Comment:**
The authors have addressed many, but not all, of my concerns with their revisions. Specific comments are below, but I would highlight a) that while it should not preclude publication, I would strongly encourage the authors to update their introduction to better motivate the importance of their study, and more importantly b) the discussion of adjustments to the Twomey effect is still lacking.
**Reply to General Comment:**
We thank the Referee for this comment and revised the introduction and included a discussion on the adjustments to the Twomey effect.

**Comment 1:**
1. Line 35: It's not clear that this is the minimum value, not the diurnally-averaged value. I would suggest that the diurnally-averaged value is the more appropriate one to report and is easier to square with your assertion that the effect is "non-negligible" yet still small.
**Reply 1:**
Thank you for the suggestion to rather present the diurnally-averaged values for instantaneous net radiative forcing and -heating rates. For a better understanding, we now list both, the diurnally-averaged and the noon values in the revision and apply the following changes:

[revised manuscript text omitted]

**Comment 3:**
3. Line 292: The "ppmv" must be a typo in Haywood et al. (2021), CO values were nowhere near that high even in the densest plumes. From the Haywood paper, my interpretation is that Figure 17 (showing the ~75 ppbv mode of CO) is from the boundary layer, measured at Ascension Island. Values of several hundred ppbv were more common in free tropospheric plumes during ORACLES and CLARIFY. Perhaps it would be good to take a step back and reconsider what the purpose of this CLARIFY comparison is in the first place?

**Reply 3:**
We also consider the "ppmv" mentioned by Haywood et al. (2021) in Figure 17 to be a typo, so we have revised the classification of the individual CO mixing ratio measurements in the PBL between DACCIWA and CLARIFY as follows:

Lines 263-266 (revised ms):
CO mixing ratios between 60 ppbv and 160 ppbv within the Southern Atlantic maritime boundary layer as measured on Ascension Island during the CLARIFY-2017 campaign (Haywood et al. 2021), show comparable CO enhancement as the DACCIWA measurements, for periods within biomass burning plumes advected from Central Africa.

**Comment 4:**
4. Lines 354-363: The new additions really aren't responsive to my main concern, which is that adjustments in terms of liquid water path and cloud fraction can substantially offset or enhance the Twomey effect. To just give a few recent high profile examples, Toll et al. (2019) found that liquid water path adjustments tend to offset ~1/3 of the Twomey effect in pollution tracks whereas Manshausen et al. (2022) found that liquid water path increases greatly enhance cooling under weaker inversions. Meanwhile, Chen et al. (2022) found that cloud response to an effusive volcanic eruption was dominated by cloud fraction increases. Wall et al. (2022) also found that the cloud fraction effect rivals or exceeds the Twomey effect (technically Twomey effect combined with liquid water path response) globally when accounting for meteorology.

To be honest, the "negative Twomey effect" literature is unpersuasive to me (you really need very high aerosol concentration for a decrease in the absolute number activated as opposed to decreased sensitivity of increases) and I'm not sure how it's relevant to your paper, as you measured increases in drop number/decreases in droplet size for a relatively moderate aerosol enhancement.

**Reply 4:**
We thank the referee for pointing out the relevant studies and want to include the discussion of adjustment mechanisms in the revised manuscript. We removed the discussion of an anti-Twomey effect and discuss the saturation effect as described by Wang et al. (2015).

Lines 332-349 (revised ms):

The origin and long-range transport of aerosol, as is the case in West Africa could play a role for the relationship between aerosol number concentration and the ED of clouds (e.g. Panicker et al., 2010). A saturation of the Twomey effect has been observed at AOTs of 0.4 to 0.5 and above by Wang et al. (2015). A mean AOT of 0.38 from the Aeronet data in the campaign region might explain the comparatively small difference in effective diameters between polluted and less polluted low-level clouds in our study.

Toll et al. (2019) found in pollution tracks in various regions around the globe that aerosols generally increase cloud brightness, mainly due to the Twomey effect. An increase of the liquid water path (LWP) as a result of cloud adjustment would be cancelled out by an entrainment effect, which leads to an overall reduced sensitivity of LWP towards anthropogenic emissions. This study shows that a decrease of the LWP can offset the Twomey effect by 23 %.

Wall et al. (2022) suggest that LWP is reduced by increases in sulfate aerosol, accompanied by a delay or suppression of precipitation. Also, Pante et al. (2021) find a correlation between increased anthropogenic aerosol emissions and reduced rainfall during the DACCIWA campaign.

Another study by Manshausen et al. (2022) found a significant negative forcing as a result of an increase of the LWP in ship tracks. The presence of a strong inversion opposes this effect, leading to the assumption that a deepening of clouds is necessary for an LWP amplification.

The pollution effect identified in our study, without being able to distinguish between various contributions to cloud adjustments, is used as a basis to derive the instantaneous cloud radiative forcing and the instantaneous heating rates based on greater CDNC and smaller ED.

**Further changes:**

Line 68, 226: Changed Ivory Coast to Côte d'Ivoire

Lines 260-263 (revised ms):

Figure 4: Correlation of accumulation mode aerosol and CO mixing ratio from all measurements <1800 m of the DLR Falcon with a linear best fit of $N_{cacc-aerosol}$= -248.05 ± 2.65 + 2.83 ± 0.02 · CO [ppbv]. The CO mixing ratio hence is used as a proxy for the degree of pollution and the abundance of activated cloud condensation nuclei within low-level clouds. Henceforth clouds with CO levels ≤135 ppbv (22nd percentile) and ≥155 ppbv (79th percentile) are characterised as less polluted or substantially polluted, respectively.